# Divergent vertebral formulae shape the evolution of axial complexity in mammals

Yimeng Li[1,2], Andrew Brinkworth [1], Emily Green[3], Jack Oyston [1], Matthew Wills[1]✉ & Marcello Ruta [3]✉

Complexity, defined as the number of parts and their degree of differentiation, is a poorly explored aspect of macroevolutionary dynamics. The maximum anatomical complexity of organisms has undoubtedly increased through evolutionary time. However, it is unclear whether this increase is a purely diffusive process or whether it is at least partly driven, occurring in parallel in most or many lineages and with increases in the minima as well as the means. Highly differentiated and serially repeated structures, such as vertebrae, are useful systems with which to investigate these patterns. We focus on the serial differentiation of the vertebral column in 1,136 extant mammal species, using two indices that quantify complexity as the numerical richness and proportional distribution of vertebrae across presacral regions and a third expressing the ratio between thoracic and lumbar vertebrae. We address three questions. First, we ask whether the distribution of complexity values in major mammal groups is similar or whether clades have specific signatures associated with their ecology. Second, we ask whether changes in complexity throughout the phylogeny are biased towards increases and whether there is evidence of driven trends. Third, we ask whether evolutionary shifts in complexity depart from a uniform Brownian motion model. Vertebral counts, but not complexity indices, differ significantly between major groups and exhibit greater within-group variation than recognized hitherto. We find strong evidence of a trend towards increasing complexity, where higher values propagate further increases in descendant lineages. Several increases are inferred to have coincided with major ecological or environmental shifts. We find support for multiple-rate models of evolution for all complexity metrics, suggesting that increases in complexity occurred in stepwise shifts, with evidence for widespread episodes of recent rapid divergence. Different subclades evolve more complex vertebral columns in different configurations and probably under different selective pressures and constraints, with widespread convergence on the same formulae. Further work should therefore focus on the ecological relevance of differences in complexity and a more detailed understanding of historical patterns.

Biological complexity[1], taxonomic diversity[2,3] and morphological disparity[4,5] are three fundamental components of macroevolutionary dynamics. Increasingly, diversity and disparity have been examined alongside each other, especially in analyses of extinct organisms[6–10]. In contrast, biological complexity remains remarkably understudied. Measuring complexity is a more challenging prospect than quantifying either diversity or disparity[11] but has vast potential for illuminating the origin of body plans[12,13], the imbalances in species richness across groups[14] and the temporal and group-specific patterns of morphological diversification[15–17].

It has long been recognized that biological complexity can be indexed at various levels and that changes across levels are often decoupled, such that one is a poor predictor of the others[18]. In its simplest formulation, complexity is defined as the number of constituent parts, or types of parts, in an organism (for example, genes, cells, tissues and organs). This definition is the most common in empirical studies of complexity, partly because of its immediacy but also because it translates into simple indices[19–22]. The hierarchical organization of biological systems offers additional proxies for complexity, such as the length and interconnectedness of biochemical pathways and gene regulatory networks or the degree of integration and modularity in the form and function of organismal parts[23]. One key aspect of anatomical complexity is the proliferation of, and the differentiation between, serial homologues[23–30]. The complexity of serial structures can be quantified as the number of elements forming a series (for example, 24 presacral vertebrae), the number of element types (for example, three types of presacral vertebrae: cervical, thoracic and lumbar) and/or the number of elements of each type (for example, 7 cervicals, 12 thoracics and 5 lumbars). Furthermore, these numbers can be synthesized as summative indices of the relative abundance and distribution of element types, analogous to the diversity of species across communities or individuals within species in an ecological sample[21,25,26]. Lastly, the morphological complexity of serial structures can be quantified, inter alia, as the total range of variation among elements, the sum of differences between sequentially adjacent elements and the direction and magnitude of gradients and slopes along a series[24,28,29].

Throughout the history of life, there has been an undisputable net increase in diversity[31], disparity[32] and maximum complexity[33]. However, as with other empirical evolutionary rules (for example, Cope–Depéret Rule of increasing body size over time[34–37]), the dynamics of increases in complexity are unclear and evidence for their generality is equivocal[38]. At one extreme, the 'zero-force evolutionary law' (ZFEL) states that in the absence of drivers or constraints, mean complexity tends to increase over time[22,39]. This could happen because increasing numbers of serial homologues or gene copies increase the number of degrees of freedom available as the substrate for differentiation. Unless otherwise constrained by anatomical, functional, genetic or developmental links, these components may passively differentiate with time. Natural selection may facilitate or prevent this divergence but will operate either in addition or in opposition to the ZFEL[23–25,40–43]. Increases in mean and maximum complexity can result from the balance between three, mutually non-exclusive processes often operating simultaneously and potentially in opposite directions across subclades[35,44,45]. (1) Passive processes ('diffusive' evolution) imply that evolving lineages undergo random walks but the extent of decreases is limited by the existence of lower bounds, such as biomechanical or physiological constraints. (2) Clade sorting involves the preferential radiation of clades with higher intrinsic complexity and the extinction of those with less. (3) Driven processes point toward more frequent increases than decreases, such that minimum, mean and maximum values move progressively farther away from the initial lower bound[1,18,22,25,44,45]. Driven trends have a particular conceptual importance because individual lineages can be viewed as independent statistical replicates of the evolutionary process, such

that parallel increases in multiple lineages can be taken as evidence for an underlying evolutionary tendency[22,25,44,45]. So far, support for a driven trend in increasing complexity through time has only been found in an overarching study of crustacean tagmosis[25], where complexity was indexed as a function of the number of paired appendages in different morphofunctional categories. However, no analyses with a comparable taxonomic reach exist for other groups.

Here, we use a similar methodological and conceptual approach to examine complexity in another exemplary model system, the vertebral column of mammals[28–30], although other serial structures (for example, body segments, paired appendages and teeth) would be suitable in other groups (for example, annelids, arthropods and vertebrates)[24–27]. The plasticity of the mammalian column results from developmental[46–51], ecological[47,52–54], functional[51,55–57] and evolutionary[48,49,53,54,58,59] factors, which also explain differences in vertebral numbers. For example, thoracic and lumbar counts are relatively conserved in some groups (for example, Marsupialia, 'Artiodactyla' and Felidae) but vary in others (for example, Afrotheria, Cetacea, Chiroptera, Primates and Xenarthra)[46–49,52,53,60–65]. Recent studies have examined axial regionalization near the origin of crown mammals and in several extant clades[28,29,53–59,66]. Furthermore, inferred ancestral conditions for vertebral counts have been mapped onto phylogenies to reconstruct major shifts in homoeotic domains[48,49,52,53,56,67].

In this article, we test for trends in the evolution of presacral complexity and investigate whether any such trends arose by passive or driven processes. We address three integrated hypotheses: $H_0 1$, vertebral counts do not differ significantly between major mammal groups; $H_0 2$, changes in complexity are non-directional; and $H_0 3$, rates of change do not exhibit departures (shifts) from an initial 'background' rate (namely, the evolutionary rate inherited by all branches of a phylogeny under a Brownian motion model of trait evolution). To address these hypotheses, we calculate several indices of complexity from the presacral formulae of a large and diverse sample of extant species and estimate index values at the internal nodes of a time-scaled phylogeny[68]. We use these estimates to quantify the magnitude and directionality of changes along ancestor–descendant lineages and to analyse temporal and group-specific patterns of axial 'complexification'[22,35,44,66]. We measure the relative incidence of passive versus driven processes on trends by examining how the total skewness of each index is partitioned across groups[45]. Lastly, we model the evolution of complexity under a relaxed-clock Brownian motion[69,70] process to detect branch-specific shifts in rates of change.

## Complexity indices and reference phylogeny

Two complexity indices derived from information theory and adapted from metrics used in ecology and archaeology[21] are applied to the presacral and thoracolumbar sections of the vertebral column (Fig. 1) of 1,136 extant species (Supplementary Data 1 and 2). As the number of cervical vertebrae is nearly constant in mammals, we discuss primarily the thoracolumbar section, unless specified otherwise. The Brillouin index of numerical diversity ($H_R$) and the evenness index of proportional distribution ($H'_R$)[25–27,71–75] quantify complexity in terms of the relative numerical richness of vertebrae in each column region and the degree to which they are apportioned uniformly across regions, respectively. Furthermore, we use presacral (CTL), thoracolumbar (TL), thoracic (T) and lumbar (L) counts, as well as thoracic to lumbar (T:L) ratios (both unstandardized and logit-transformed), as additional indices that describe region-specific elongation and the relative sizes of the thoracic and lumbar homoeotic domains[48,53,76]. We use a taxonomically pruned version of a recent phylogeny[68] (Fig. 2 and Supplementary Data 3), with species assigned to ten groups: Afrotheria, Cetartiodactyla, Chiroptera, Eulipotyphla, Glires, Monotremata + Marsupialia, Perissodactyla, Ferae, Euarchonta and Xenarthra. Except for Monotremata + Marsupialia (a paraphyletic group to accommodate the small number of monotremes in our taxon sample), all other groups are clades.

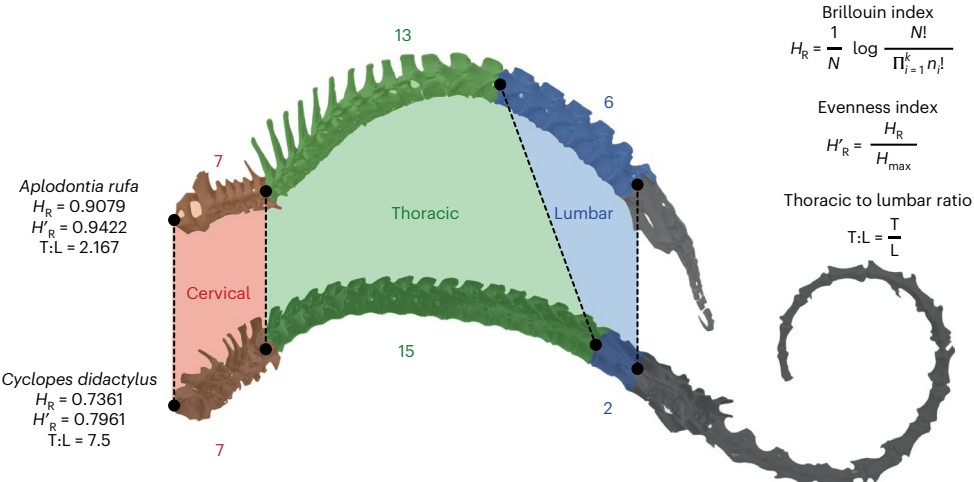

**Fig. 1 | Examples of mammalian vertebral columns and calculations of three complexity indices.** The vertebral columns of a mountain beaver (*Aplodontia rufa;* specimen FMNH 57831; Field Museum of Natural History, Chicago, USA) and a silky (or pygmy) anteater (*Cyclopes didactylus;* specimen UMZC E621; University Museum of Zoology, Cambridge, UK) are shown with colour-coded presacral regions. For each column, we report the presacral Brillouin and evenness indices and the T:L ratio. Both columns courtesy of Elizabeth Griffiths (University of Oxford, UK; three-dimensional rendering from computerized tomography scans) and Roger Benson (University of Oxford, UK); final figure assembly courtesy of Olivia Wills.

## Results

### Vertebral counts differ significantly between groups

For each of the CTL, TL, T and L counts, Poisson regression analyses[77] find significant differences in vertebral numbers between groups. Goodness-of-fit tests (analyses of deviance) indicate that all regression models depart significantly from a null model ($H_0$ = predicted group-specific counts identical to observed counts) (Supplementary Table 1). Out of 45 pair-wise group comparisons, post hoc Tukey tests retrieve 18 significant differences between group-specific counts with TL, 12 with CTL, 16 with T and 15 with L (Supplementary Table 1). Cetartiodactyla, Perissodactyla, Afrotheria, Chiroptera, Glires and Xenarthra are the most frequently occurring clades in the total number of significant pair-wise comparisons across all counts. All 12 significant comparisons with CTL are also retrieved with TL, 10 are shared between TL and T, 7 between TL and L and 2 between T and L. Afrotheria and Perissodactyla are the most widely represented clades in significant pair-wise comparisons of T counts, while Xenarthra and Cetartiodactyla feature prominently in comparisons of L counts. Finally, the only two significant comparisons in common to T and L involve Cetartiodactyla and each of Perissodactyla and Afrotheria (Supplementary Table 1 and Supplementary Information).

### Complexity is distributed unevenly across groups

For the CTL and TL Brillouin and evenness and the untransformed and logit T:L, the total range of values is apportioned unevenly across groups, with varying degrees of overlap between group-specific index values and with differences in the number and location of distribution modes (Fig. 3 and Supplementary Data 1). Spearman's rank-order correlation tests show that the group-specific index ranges do not correlate significantly with group size (Supplementary Data 1) and phylogenetic analyses of variance[78] return no significant differences in mean index values between groups (Supplementary Table 2). Monotremata + Marsupialia, Afrotheria and Xenarthra encompass the largest proportions of the total range of values for CTL Brillouin, CTL and TL evenness and T:L. Monotremata + Marsupialia and Afrotheria, along with Cetartiodactyla, also span much of the total range for TL Brillouin, while Monotremata + Marsupialia, Cetartiodactyla and Glires occupy a large proportion of the total range for logit T:L. In contrast, Ferae and Perissodactyla include the smallest proportions of the total range for most indices (Supplementary Data 1). In most groups, the indices are either bimodally or multimodally distributed (Hartigans' dip tests; $H_0$, unimodal distributions)[79], suggesting distinct evolutionary trajectories (for example, trends towards separate optima) but in three early diverging groups—Monotremata + Marsupialia, Afrotheria and Xenarthra—they are unimodal (Fig. 3 and Supplementary Data 1).

### Changes in complexity are concentrated in younger branches

We visualize changes in complexity indices across the phylogeny using continuous trait mapping[80] (Fig. 4 and Extended Data Fig. 1), with colour-coded maximum likelihood estimates at internal nodes and interpolated values along branches. We focus on TL (Fig. 4a), TL Brillouin and evenness (Fig. 4b,c) and untransformed T:L (Fig. 4d). The distributions of CTL Brillouin and evenness (Extended Data Fig. 1a,b) and logit T:L (Extended Data Fig. 1c), resemble their counterparts in Fig. 4 and are not discussed.

In agreement with previous findings[49], a TL count of 19 is estimated at the deepest nodes of the phylogeny (for example, separation between Marsupialia and Placentalia; emergence of major placental cohorts) and near the roots of most extant orders. Small deviations from this plesiomorphic value characterize Chiroptera with an estimated ancestral count of 18 (19 in ref. [49]), Carnivora with 20 (see also ref. [46]) and Perissodactyla and Afrotheria with 22 each (22–24 in Perissodactyla and 21–30 in Afrotheria in ref. [46]). Within Afrotheria, 22 is also the most likely ancestral estimate for the ecologically diverse Paenungulata (Sirenia, Proboscidea and Hyracoidea) (23 in ref. [49]). A TL count of 19 is also probably plesiomorphic for primates, despite conspicuous differences in vertebral counts in this group[49,81]. At less deep nodes, TL estimates are somewhat variable. This is a function of remarkable differences in vertebral counts in small groups of closely related species and in large subclades within some orders (for example, Chiroptera and Cetacea; Fig. 4a and Supplementary Data 1).

Along the deepest branches, changes in complexity near the roots of sister clades reveal consistent patterns for all information theory indices. The branches subtending Monotremata and Theria (Marsupialia + Placentalia) feature a decrease and an increase, respectively. Increases occur along the branches leading to Marsupialia and Placentalia. Within Placentalia, Atlantogenata (Xenarthra + Afrotheria) show a decrease, while their more speciose sister clade, Boreoeutheria,

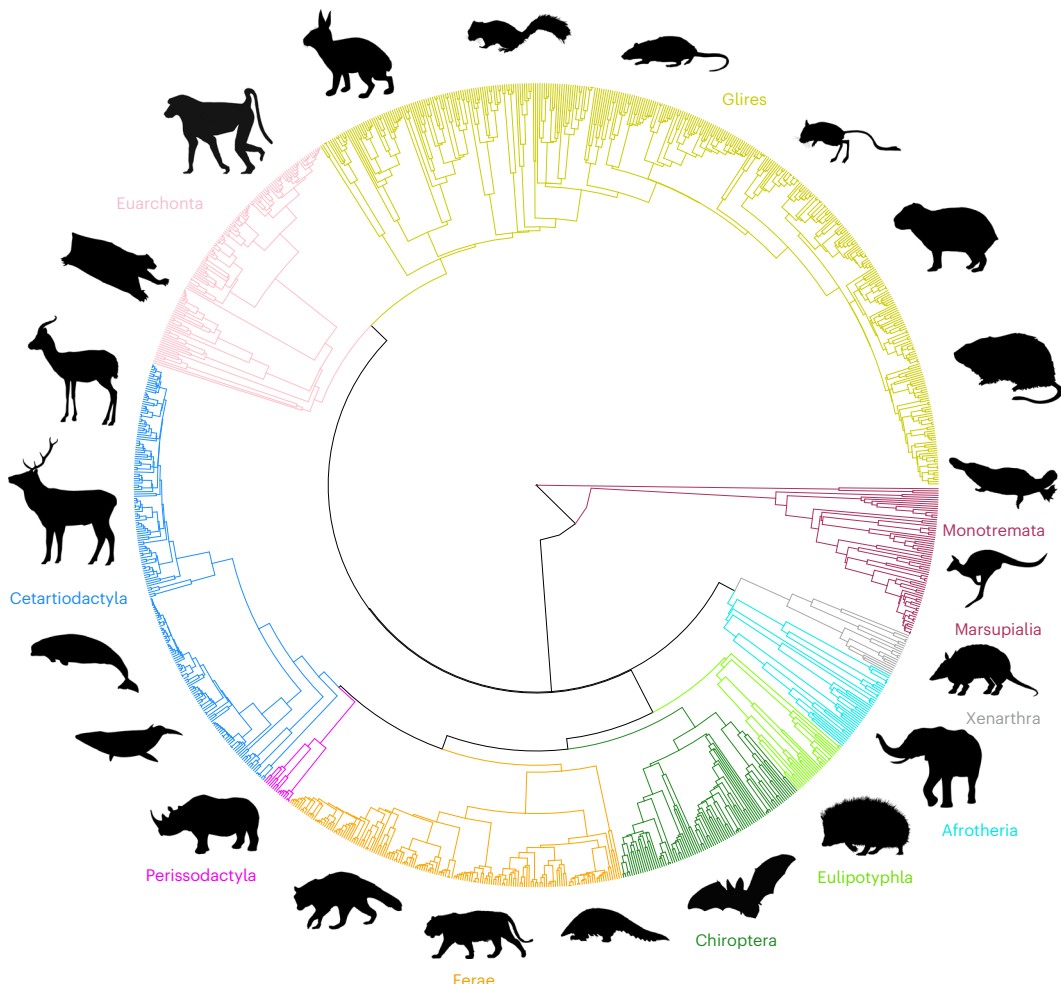

**Fig. 2 | Time-calibrated phylogeny of 1,136 extant mammal species used in this study.** Major groups are indicated by different colours and symbolized by black silhouettes of representative taxa (not to scale). Icon credits: except for the capybara (A.B.), the silhouettes are sourced from PhyloPic.org. Individual creator credits: hedgehog, Inessa Voet; black rat, Ferran Sayol; northern three-toed gerboa and raccoon, Margot Michaud; hispid cotton rat, Natasha Vitek; Siberian chipmunk, Nina Skinner under CC BY 3.0; yellow baboon, Owen Jones; mountain gazelle, Rebecca Groom under CC BY 3.0; African elephant and red kangaroo, Sarah Werning under CC BY 3.0; red deer, beluga whale, European rabbit, black rhinoceros, tiger, Philippine pangolin, nine-banded armadillo and duck-billed platypus, Steven Traver; Sunda colugo and Townsend's big-eared bat, Yan Wong.

features an increase. Increases also occur near the roots of the two major boreoeutherian clades—Laurasiatheria and Euarchontoglires—as well as along the basal branches of Euarchonta and Glires. Within Laurasiatheria, Eulipotyphla show a decrease, while other laurasiatherians (Scrotifera) experience an increase. At the next deepest node, Chiroptera show a decrease whereas remaining Scrotifera (Ferae + Perissodactyla + Cetartiodactyla) exhibit an increase (for an account of group-specific patterns, see Supplementary Information).

### TL counts correlate negatively with T:L ratios

A phylogenetically independent contrasts[82] analysis yields a significant negative correlation between TL counts and T:L ratios across all sampled species (Supplementary Table 3). As with other indices, the ratios reveal a markedly heteroskedastic distribution (Supplementary Fig. 1), with a little over 2% of variance in the data explained by the regression model. A locally estimated scatterplot smoothing (LOESS) curve fitted to the T:L versus TL scatterplot (Fig. 5a) shows a steep negative slope for TL up to 19, a negligible rise to 20 and a gentle negative slope throughout higher TL counts (≥20). The dispersion of T:L values increases rapidly at TL ≥16 but reduces drastically at TL ≥29, with the long tail at the right-hand side of the distribution reflecting the large number of lumbar vertebrae in many Cetacea[57,62] (Fig. 5b,c and

Supplementary Data 1; for the correlation between T:L and each of T and L, see Supplementary Information).

### TL counts correlate positively with complexity

TL counts are positively correlated with each of the TL Brillouin and evenness indices, although significantly so only for TL Brillouin (Supplementary Figs. 2 and 3 and Supplementary Table 3; see Supplementary Information and Supplementary Figs. 4 and 5 for patterns associated with CTL). This suggests that mammals with increasingly more elongate postcervical regions show a tendency for thoracic and lumbar elements to be distributed equitably. This pattern is rendered more elaborate by variations in T and L, whereby mammals with lower T and/or higher L also reveal greater relative numerical richness (Supplementary Information).

### Thoracolumbar complexity and domains

Phylogenetic generalized least square regressions[83] support a strong, negative and highly significant correlation between T:L ratios and each of the TL evenness and Brillouin indices (Supplementary Figs. 6 and 7 and Supplementary Table 3). The seemingly linear decrease in TL Brillouin and evenness values with increasing T:L ratios is not consistent among groups and reversals (especially for TL Brillouin) occur in

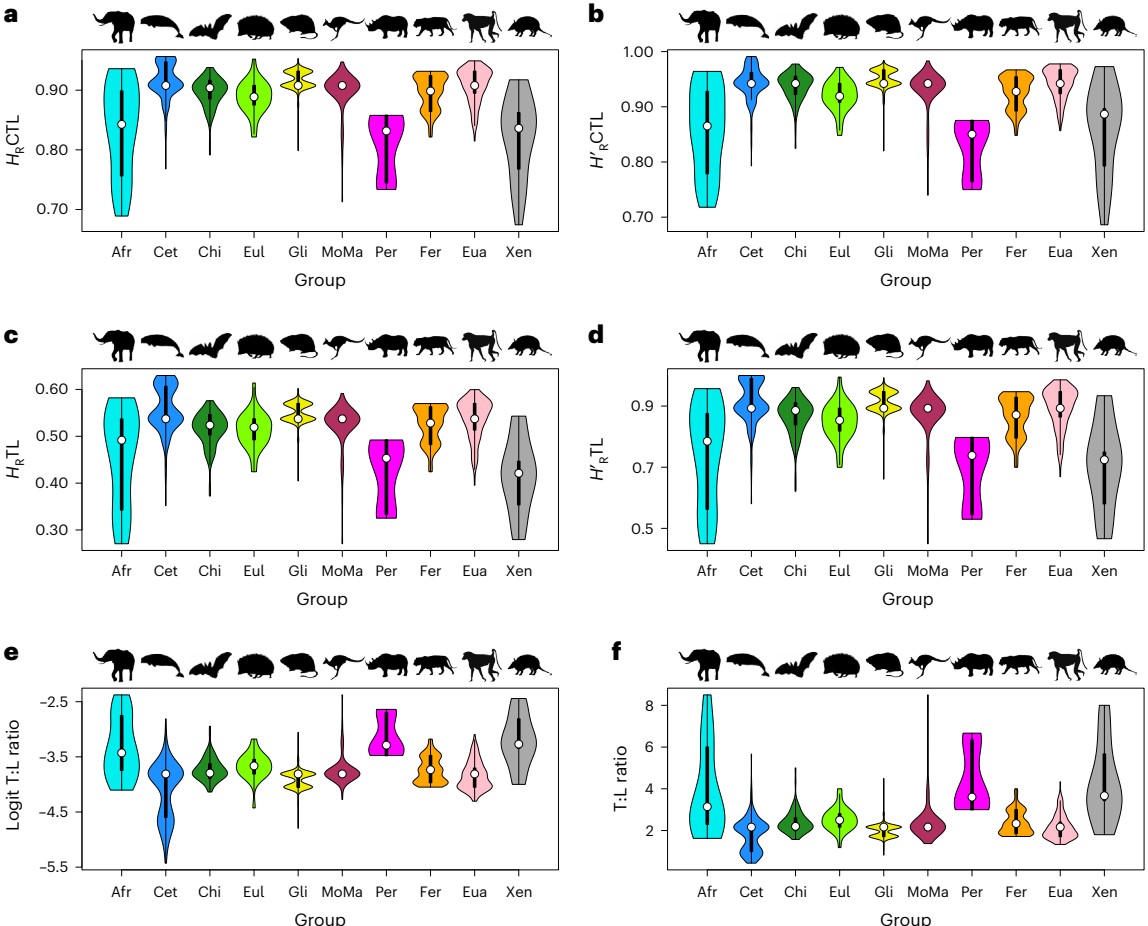

**Fig. 3 | Violin plots.** Distributions of six complexity indices colour-coded by group. Data are presented as probability density distributions (violin outlines), median values (white circles) and interquartile ranges (solid black vertical bars). The groups are symbolized by black silhouettes, as in Fig. 2. **a,b**, Presacral Brillouin (**a**) and evenness (**b**) indices. **c,d**, Thoracolumbar Brillouin (**c**) and evenness (**d**) indices. **e,f**, Logit-transformed (**e**) and unstandardized (**f**) T:L ratio. Afr, Afrotheria; Cet, Cetartiodactyla; Chi, Chiroptera; Eul, Eulipotyphla; Gli, Glires; MoMa, Monotremata + Marsupialia; Per, Perissodactyla; Fer, Ferae; Eua, Euarchonta; Xen, Xenarthra. Image credits for mammal silhouettes are as in Fig. 2.

Afrotheria, Cetartiodactyla, Euarchonta and Glires (Supplementary Figs. 6 and 7).

**Changes in complexity are both directional and sustained**

To establish whether changes in complexity conform to trends and whether any such trends are predominantly passive or driven, we implement descendant–ancestor tests[84–86] and subclade tests[45,84,87–90]. For the descendant–ancestor tests, we first examine temporal patterns of complexity change by regressing maximum likelihood estimates of the complexity indices at the internal nodes of the phylogeny (ancestral node values) against node ages. Subsequently, we test whether high initial levels of complexity bias downstream changes towards increases in daughter lineages by regressing the differences between successive node estimates (descendant value minus ancestor value for each branch) against ancestral node values. We model all correlations using robust linear regression, which is relatively insensitive to outliers and heteroskedasticity[91].

Heteroskedasticity-robust Wald $F$-tests[92] show that the slopes and intercepts of most regression models differ significantly from zero (Supplementary Table 4). The ancestral values of the CTL and TL Brillouin and evenness indices correlate positively (but not strongly) with node ages, both across phylogeny (Fig. 6a,c, Extended Data Fig. 2a,c and Supplementary Table 4) and in most individual groups (Supplementary Figs. 8–11 and Supplementary Table 4). The only clades

exhibiting a temporal trend of decreasing complexity are Afrotheria (CTL Brillouin; CTL and TL evenness), Xenarthra (TL Brillouin) and Perissodactyla (CTL and TL Brillouin and evenness) but in none of them is the regression slope significant for any index (Supplementary Table 4). For the unstandardized and logit T:L, node estimates and node ages are mostly negatively correlated (except in Perissodactyla for unstandardized and logit T:L and Afrotheria for logit T:L), indicating a tendency for ratios to decrease towards the present (that is, thoracic and lumbar vertebrae tend to be apportioned uniformly) (Fig. 6e,g, Supplementary Figs. 12 and 13 and Supplementary Table 4). This pattern is mainly associated with increasingly larger numbers of lumbar elements, primarily within Cetacea and in several Afrotheria and primates.

For all indices, the correlations between ancestral values and descendant–ancestor differences, both across the entire taxon set and in most major groups, are positive, although never strong. Except in Eulipotyphla (for all indices), Afrotheria and Xenarthra (for logit T:L) and Afrotheria (for TL Brillouin), the slopes of the regression models are mostly significant (Fig. 6b,d,f,h, Extended Data Fig. 2b,d, Supplementary Figs. 14–19 and Supplementary Table 4), suggesting that, as complexity values at nodes become larger, so do the differences between adjacent nodes[93]. More broadly, for all information theory indices, increases significantly outnumber decreases (two-tailed sign tests) across phylogeny. Furthermore, the mean magnitude of combined increases and decreases is positive and significantly different

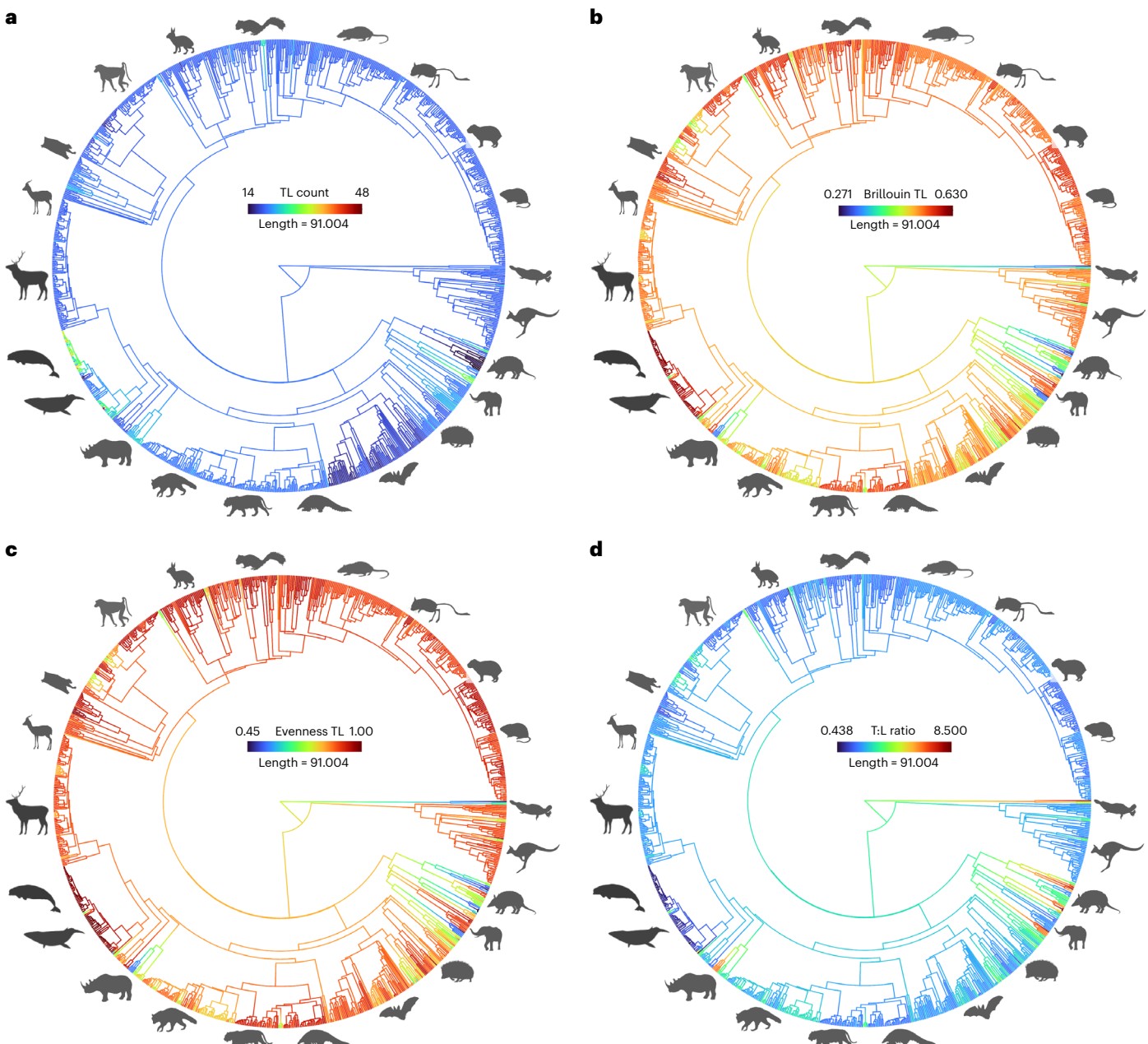

**Fig. 4 | Continuous mapping of complexity indices across the phylogeny.** The values of four complexity indices are mapped onto the phylogeny using colour gradients. Index values at the internal nodes are estimated through maximum likelihood and those along the branches are interpolated between the nodal estimates. For each index, the colour scales range from its minimum to its maximum value. The lengths of the scale bars are in millions of years. **a**, Thoracolumbar count. **b,c**, Thoracolumbar Brillouin (**b**) and evenness (**c**) indices. **d**, Unstandardized T:L ratio. Image credits for mammal silhouettes are as in Fig. 2.

from zero (two-tailed, one-sample Wilcoxon tests), with the mean magnitude of increases greatly exceeding the mean absolute magnitude of decreases (two-tailed, unpaired two-sample Wilcoxon tests) (Supplementary Table 4). For the unstandardized T:L, decreases outnumber increases, indicating a tendency for thoracic and lumbar elements to attain similar proportions.

In addition to the tests above, we quantify subclade skewness as a partial test of a driven trend[45]. The skewness of a continuous trait (in particular, a right-skewed trait) within a parent clade suggests a hard bound in the opposite direction to the skew, with diffusion away from this bound. Where subclades within the parent clade are also skewed in a similar direction, this suggests a replicated and driven trend (note: the mean values of the subclades may be distributed symmetrically around the parent clade's mean or may occur further in the direction

of the overall skew[45]). Formally, large contributions of within-group skewness (SCW) indicate that trends are predominantly shaped by driven processes, while large contributions of between-group (SCB) and heteroskedasticity-related (SCH) skewness suggest the prevalence of passive processes[45]. In the case of the unstandardized T:L, a subclade test shows that slightly over 32% of its total skewness is accounted for by SCW, a little over 50% by SCH and the rest by SCB (Supplementary Table 5). For other indices, the proportion of SCW varies (32% for TL Brillouin, 29.9% for TL evenness, 26.5% for CTL Brillouin and 24.6% for CTL evenness) and accounts for the second largest proportion of total skewness after SCH (Supplementary Figs. 20–24 and Supplementary Table 5). At the broad taxonomic scale used here, these results indicate that passive processes are primarily responsible for trends in axial complexity but that driven components of change are also apparent.

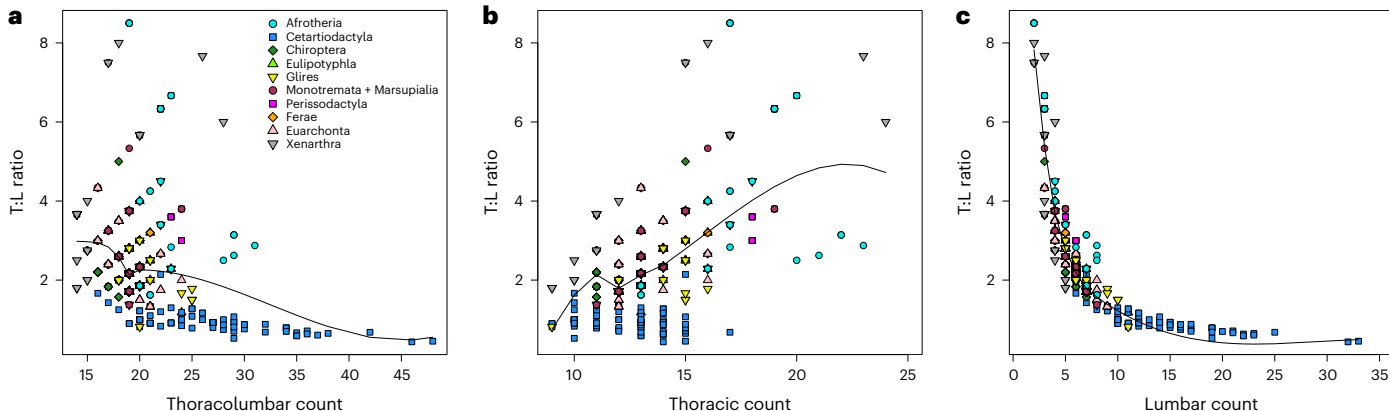

**Fig. 5 | Patterns of thoracolumbar differentiation across major groups.**
**a–c**, Bivariate scatterplots of T:L ratios versus thoracolumbar (**a**), thoracic (**b**) and lumbar (**c**) counts, with LOESS regression curves superimposed over each plot. Mammal groups are identified by distinct colours and symbols. A variant of the plot in **a** is also reported in Supplementary Fig. 1, where the distribution of T:L ratios is represented by a box and whisker plot.

The limited contribution of SCW may be attributed to various factors[45]. For instance, the variance of a trait may differ across subgroups or correlate negatively with their size. In addition, a trait may appear skewed in opposite directions in different subgroups and its distribution in one or more of these may fall outside the right-hand tail of the parent clade's distribution. Lastly, large subgroups may obliterate smaller-scale patterns in the distribution of total skewness[45]. In the Supplementary Information, we comment briefly on the influence of group size on the distribution of skewness using the TL Brillouin and evenness indices as examples (Extended Data Fig. 3).

### Rate shifts reveal divergent patterns between clades

We find support for a multiple-rate model of complexity change. All complexity indices are characterized by widespread rate shifts with broadly congruent branch locations, directions, magnitudes and posterior probabilities. We discuss rate distribution mostly in relation to TL Brillouin (Extended Data Fig. 4a) and evenness (Extended Data Fig. 4b) (for patterns associated with other indices, see Extended Data Fig. 5). Shifts are absent or negligible in most deep branches, indicating few departures from background rates. In contrast, many of the more apical branches reveal shifts with large posterior probabilities, mostly representing rate decreases. Although increases are similarly widespread, they feature mainly at, or near, the tips of the phylogeny and are linked to divergent vertebral formulae occurring in single species and/or small subclades within the more speciose groups. Most of the branches subtending major mammal cohorts and orders are not underpinned by rate shifts, except for Atlantogenata (Afrotheria + Xenarthra) and Scandentia (tree shrews) which feature, respectively, an increase (for both indices) and a decrease (for TL evenness) with large posterior probabilities. For both indices, two conspicuous decreases occur in Marsupialia, one near the base of a diverse clade of small to mid-sized omnivores and carnivores (Tasmanian wolf, quolls, numbats, bilbies and bandicoots), the other near the base of macropods (bettongs, kangaroos, potoroos and wallabies). Additional decreases, usually with large posterior probabilities and typically located near the terminal branches, feature in all other groups. Notable examples of such decreases (involving TL Brillouin and/or evenness) characterize Chiroptera (mouse-eared bats; TL evenness), Ferae (weasels, ferrets, mink and their allies, felids, canids and pinnipeds), Cetacea (various mesoplodont beaked whales; TL evenness), 'Artiodactyla' (Old World deer and several bovid clades, such as Caprini and Antelopini), primates (bamboo, ruffed and true lemurs, leaf-eating monkeys and macaques), Lagomorpha (rabbits and hares) and Rodentia (early diverging groups, such as several flying, ground and tree squirrels within sciurids and many clades in more deeply nested rodent suborders) (Extended Data Fig. 4).

In the case of CTL evenness (Extended Data Fig. 5b), as well as the unstandardized and logit T:L (Extended Data Fig. 5c,d), complex patterns of rate shifts occur in Carnivora, Cetartiodactyla and Chiroptera. For those three indices, Carnivora show interested increases and decreases with low to moderate posterior probabilities along the basal branches separating the major clades of Caniformia and (for T:L only) Feliformia. Interested increases and decreases in T:L with moderate to high posterior probabilities also occur in Chiroptera (evening bats). Uniquely among all tested indices, T:L shows small interested increases within Perissodactyla[94] (one at the base of the entire clade, the other at the base of Rhinocerotidae) (Extended Data Fig. 5c). In Cetacea, interspersed increases and decreases characterize CTL Brillouin and evenness (Extended Data Fig. 5a,b), whereas widespread increases are associated with the unstandardized and logit T:L (Extended Data Fig. 5c,d). Increases with large posterior probabilities feature predominantly among baleen whales (Mysticeti) and dolphins, whereas decreases occur near the roots of toothed whales (Odontoceti).

## Discussion

The strikingly divergent thoracolumbar patterns of five clades—Cetacea, Afrotheria, Xenarthra, Carnivora and Chiroptera[95–104]—have broad evolutionary and ecological relevance to our understanding of axial regionalization and are therefore discussed in some detail. With the largest variance in L counts of all groups and the third largest variance in T counts after those of Xenarthra and Afrotheria (Supplementary Data 1), Cetacea rank among the mammals with the most complex thoracolumbar regions (TL Brillouin, 0.5775–0.6298; TL evenness, 0.8911–0.997). In Cetacea, postcervical elongation varies across lineages and according to habitats and ecologies[57,62,63]. Among toothed whales (Odontoceti), beaked whales commonly have 9 or 10 (rarely, 11) thoracic vertebrae, whereas in other odontocete families, as well as in baleen whales (Mysticeti), T counts ≥12 are common. Appreciably more striking are the variations in L counts. For example, the stout-bodied South American river dolphin and the pygmy right whale have three lumbar vertebrae, while the slender and elongate right whale dolphins have >30. Freshwater cetaceans generally exhibit lower TL counts and their vertebral centra tend to be elongate and spool-like. In marine species, two divergent patterns emerge[63]. Higher TL counts, coupled with abbreviated and disc-like centra, are frequently observed in small taxa adapted to fast swimming and active hunting (for example, oceanic dolphins and porpoises). Biomechanically, such features both afford rigidity and stability in the trunk and enable a powerful tail beat, allowing these animals to chase swift and agile prey. In contrast, lower TL counts usually occur in mid-sized to gigantic taxa (for example,

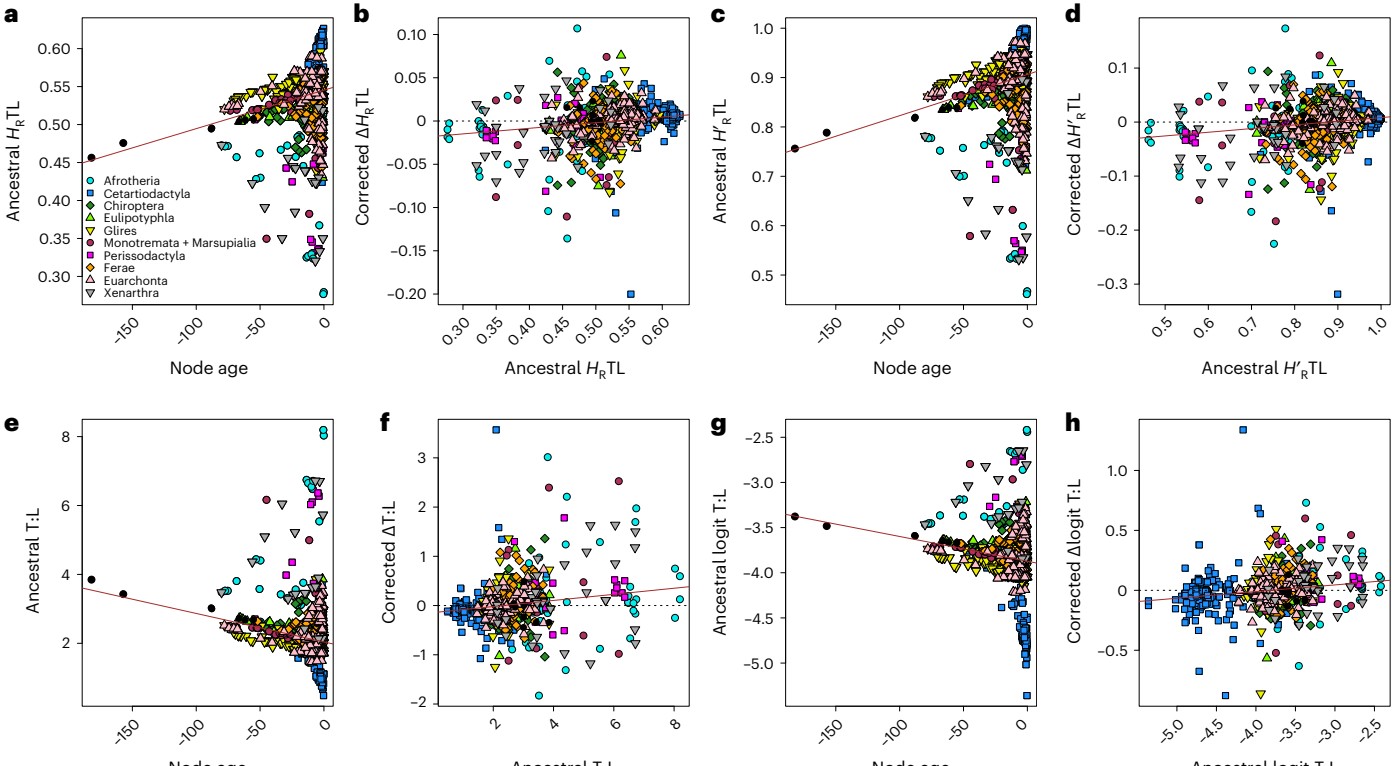

**Fig. 6 | Bivariate plots of node estimates of complexity indices versus node ages and descendant–ancestor differences versus node estimates. a–h,** Robust linear regressions between maximum likelihood node estimates of four complexity indices and node ages (**a,c,e,g**) and between descendant–ancestor differences (corrected for the regression to the mean) and node estimates (**b,d,f,h**). The brown solid lines are regression lines. The black dashed horizontal lines separate positive (increases) from negative (decreases) descendant– ancestor differences. Node ages are in millions of years, with time decreasing in the positive direction of the axis (that is, closer to the present). Mammal groups are identified by distinct colours and symbols. Black circles denote the deepest nodes of the phylogeny, corresponding to the separation between major mammal cohorts. **a,b,** Thoracolumbar Brillouin index. **c,d,** Thoracolumbar evenness index. **e,f,** Unstandardized T:L ratio. **g,h,** Logit T:L ratio.

baleen whales, killer whales, sperm whales, narwhals, belugas and beaked whales), in which variably elongate to isodimensional centra provide flexibility and manoeuvrability. Such characteristics assist these species with their vast repertoire of feeding habits, including foraging on the seabed or in discontinuous, high-density food patches in the water column[63]. During cetacean evolution, the vertebral column was released from the functional requirements associated with stance, gait and weight-bearing, all of which limit the range of vertebral numerical variation in terrestrial species[46,52,61] and became less functionally plastic and more morphologically homogeneous[57]. Alongside a simplification in the shape of the individual vertebrae, an increase in vertebral number was a key step during the transition to a fully aquatic lifestyle[62]. This transition was marked by the progressive reduction and subsequent loss of the hind limbs and a functional shift from paddling to manoeuvring in the fore limbs. At the same time, thoracic and lumbar elongation conferred greater stability to the trunk. Selection for improved stability and acquisition of a streamlined body profile may explain why numerous cetaceans exhibit similar numbers of thoracic and lumbar elements. Thus, in 48 out of 82 species in our sample, T:L varies from ~0.8 to 1.25, with absolute differences between T and L counts ranging from 0 to 2.

Unlike Cetacea, Afrotheria and Xenarthra attain trunk elongation mostly through additions of thoracic vertebrae. Afrotheria also showcase major differences in L counts, with the second largest variance after Cetartiodactyla. The plasticity of the vertebral column in Afrotheria reflects multiple biomechanical and ecological adaptations[49,52,53], exemplified by graviportal elephants, swimming manatees, burrowing aardvarks and various small to mid-sized insectivorous taxa of semi-aquatic to terrestrial habits, such as tenrecs, golden moles, otter shrews and elephant shrews. Manatees and golden moles are remarkably convergent in possessing moderately elongate thoracic regions (15, 17 or 19 vertebrae) and extremely abbreviated lumbar regions (2–4 vertebrae). Whereas in manatees these characteristics enhance column stability during swimming[95], in various golden moles they act as shock-absorbing devices, rendering the body compact while these animals probe through sand with their fore limbs and muzzle[96]. Hyraxes rank among the Afrotheria with the largest T (20–23) and L (7 and 8) counts. Their plump appearance and short limbs contrast with their elongate, strongly arched and flexible columns, allowing them to move rapidly on rock surfaces and trees[97]. Like manatees, elephant shrews also display relatively low TL counts (20 and 21). Similar to other mammals adapted to running or leaping[52,61], elephant shrews show restricted variation in their presacral formulae and, alongside aardvarks, feature the most complex presacral columns of all Afrotheria (CTL Brillouin, 0.9359–0.8985; CTL evenness, 0.9641–0.9277).

Similarly varied in terms of column construction are Xenarthra, characterized by their unusual vertebral articulations and a wide range of specializations[98], including osteoderms and burrowing lifestyles in armadillos (9–12 thoracics and 3–5 lumbars), fossoriality and arboreality in anteaters (15–17 thoracics and 2 or 3 lumbars) and suspensoriality in sloths (15–24 thoracics and 3 or 4 lumbars). The southern long-nosed armadillo shows the highest vertebral complexity in the group despite its low presacral count (C7, T9, L5; $H_R$ = 0.9174; $H'_R$ = 0.9727), whereas the Hoffmann's two-toed sloth features the lowest despite its higher presacral count (formula: C6, T23, L3; $H_R$ = 0.6743; $H'_R$ = 0.6862).

Hence, there is some suggestion that within Xenarthra higher vertebral complexity is driven in part by greater trunk stability associated with the evolution of a defensive armour and curling behaviours in armadillos, while lower complexity is partly the result of an expansion of the thoracic region and a concurrent reduction in the number of lumbar elements in arboreal species (various sloths)[52,61].

Two placental clades—the Carnivora and Chiroptera—stand out because of their markedly discontinuous vertebral count distributions. In Carnivora, the posterior thoracic and lumbar vertebrae are generally characterized by greater individual disparity across species and lower serial differentiation than the anterior thoracic and cervical vertebrae[55]. Such attributes are thought to facilitate the striking functional versatility of the posterior trunk region in this clade, as well as among mammals more broadly[59]. Consistent with this hypothesis is the fact that, despite a substantial degree of conservatism in TL counts[99], several Carnivora showcase highly divergent presacral formulae (for example, grisons, hyaenas, otters, seals and weasels[48,49,53,55]). The probability density distributions of their complexity indices (Fig. 3 and Supplementary Data 1) are strikingly multimodal. Both TL Brillouin and TL evenness show no fewer than four modes, each reflecting convergent patterns of axial regionalization in separate lineages. For both indices, felids, canids and herpestids (mongooses and their kin) cluster around an optimum at the right-hand side of the probability distributions. The first two groups consist of small- to large-sized, long-limbed digitigrade hunters built for strength and speed. In contrast, herpestids include small carnivore generalists with elongate bodies and tails and short limbs. Narrowly separated from this optimum is a second peak in the probability distribution, including large ambush predators, such as ursids, and the small, long-bodied and short-tailed mustelids (weasels and their kin), a diverse clade of terrestrial, arboreal, aquatic and fossorial taxa. Largely separate from the first two distribution modes is a third optimum almost exclusively dominated by pinnipeds, the thoracolumbar formulae of which differ from those of most other carnivores (T15, L5 in most species)[100]. Finally, an inconspicuous mode at the left-hand side of the distribution includes several ursids, mustelids and some hyaenids.

As in Carnivora, the distribution of complexity modes in Chiroptera is decoupled from phylogenetic clustering. Both TL Brillouin and (to a lesser degree) TL evenness are bimodal (Fig. 3 and Supplementary Data 1), with species from phylogenetically separate families clustered around each mode. Unlike axial complexity, patterns of vertebral fusion in Chiroptera are mostly phylogenetically structured[101]. This paradox may be explained by considering the construction of the vertebral column in light of recent embryological data. Vertebral fusion is ubiquitous in bats, confers stability and rigidity to the column and is a key adaptation for sustained flight[64,65]. As in other dorsostable groups (for example, suspensory and slow-climbing taxa), variations in TL counts are poorly constrained in bats[52,61]. Uniquely among mammals, bats feature a delayed onset of column ossification, with substantial morphogenetic patterning taking place in prenatal developmental stages[102]. This patterning is responsible for the specialized traits observed in the adults of several species, including varying degrees of vertebral fusion, restructuring of vertebral bony processes and remodelling of cervical elements in response to roosting habits[64,65]. A recent hypothesis for the origin of flight in bats[103] posits that the ancestral morphotype of Chiroptera was a nocturnal, insectivorous and arboreal placental with well-developed interdigital webbing. Subsequent stages in the evolution of bats included interdigital webbing expansion, the transition from arboreal to roosting habits via an intermediate suspensory phase[104] and shifts in prenatal sequence heterochrony. Such shifts may have relaxed constraints on thoracolumbar counts and altered the boundaries between the thoracic and lumbar domains, resulting in the remarkable proliferation of vertebral formulae in extant bats[65].

## Methods

### Data collection

Vertebral formulae (Supplementary Data 1) were obtained from published compendia[46–49,52,53,60,61,65] (Supplementary Data 2) and supplemented by data in their accompanying bibliography and citations, Boolean searches in Google Scholar (combining 'and/or' operators with keywords such as 'mammal(s)', 'vertebra(e)', 'formula(e)', 'thoracic', 'lumbar' and the names of individual mammal orders), as well as synopses from Mammalian Species (https://academic.oup.com/mspecies). Species were sampled from as many families as possible within each order to provide adequate coverage of presacral variation. Where intraspecific variation was documented, we selected the most widely represented count (usually, ≥50% of all specimens listed in a publication; but see ref. [52]). We excluded the sacral and caudal regions because their vertebral counts were often difficult to obtain[48,49]. All analyses, tests and graphs were produced in R v.4.2.0. R codes are provided in Supplementary Data 4.

### Phylogeny construction

We used a recently assembled time-calibrated phylogeny of mammals[68] built from a Bayesian analysis of a 31-gene supermatrix coupled with fossil-based backbone relationships and divergence time estimates. We exported 1,000 randomly selected trees from the posterior distribution available at vertlife.org and stored them as a 'multiphylo' object in R. Subsequently, we chose the single tree from the random sample closest to the centroid of tree space and available in Supplementary Data 3. To this end, we first extracted pair-wise tree-to-tree distances in phangorn[105] (KF.dist function), using the branch score distance (BSD)[106] as the preferred distance metric. BSD is calculated as the square root of the sum of squared differences between the branch lengths of a given tree pair. It is preferred over the more widely used Robinson–Fould (RF) distance[107] because the latter often results in narrow value distributions, heavy index saturation and limited power to differentiate alternative tree topologies. For instance, transpositions of even a single taxon pair can yield maximal changes in RF[108]. The matrix of pair-wise BSD distances was used to calculate total summed distances between each tree and all other trees in the random sample (colSums function; base R). The tree with the smallest sum of column-wise distances from all the others was chosen for all subsequent analyses and pruned to include the species in our sample (drop.tip function in ape[109]). Before performing phylogenetic comparative analyses, we re-ordered all data tabulations to ensure that the order of taxon labels matched that of the tree (match.phylo.data function in picante[110]). The circular tree (Fig. 2) with branches coloured by group was obtained with the groupClade (colour coding) and ggtree (aesthetic mapping) functions in ggtree[111].

### Complexity indices

Counts and ratios were derived from the tabulated vertebral formulae (Supplementary Data 1) using base R. The logit T:L ratios were obtained with the logit function in car[112]. The logit values are a desirable alternative to log-transformations in that they both stabilize the variance of the ratio distribution and extend its left and right extremes, such that ratios that differ only slightly are more widely spread at the tail ends of the logit distribution. The information theory indices were calculated with the heterogeneity and evenness functions in tabula[113]. The formula for the Brillouin index ($H_R$) is:

$$H_R = \frac{1}{N} \log \frac{N!}{\prod_{i=1}^{k} n_i!}$$

where $N$ is the total number of vertebrae, $n_i$ is the number of vertebrae in the $i$th region and $k$ is the number of regions. The base of the log operator can take any value. The formula for the evenness index ($H'_R$) is:

$$H'_R = \frac{N \times H_R}{\log N! - (k-d) \times \log c! - d \times \log (c+1)!}$$

where $H_R$ is the Brillouin index and $c$ and $d$ are the integer and modulus of $N$ and $k$, respectively. The Brillouin index is appropriate for count data derived from a known collection rather than a random sample[73], accommodates variation associated with the least numerically diverse vertebral types[74] and offers an intuitive characterization of complexity, in that higher values indicate greater dissimilarity across column regions while accounting for the numerical richness of vertebral types in each. Similarly, the evenness index explains how uniformly spread the vertebrae are across regions. While related, the Brillouin and evenness indices describe numerical variation in subtly different ways and need not correlate. As an example, consider the following numerical sets, each consisting of two integers that add up to the same total count: $a = \{10, 1\}$, $b = \{9, 2\}$, $c = \{8, 3\}$, $d = \{7, 4\}$ and $e = \{6, 5\}$. The Brillouin index increases from $a$ to $e$ (0.2179, 0.3643, 0.4641, 0.5271 and 0.5577), indicating increases in the relative numerical abundance of elements across the five sets. The evenness index also increases in the same order (0.3908, 0.6531, 0.8321, 0.9451 and 1), reaching its maximum value in $e$, where the two integers contribute most equitably to their sum (note: swapping the positions of the integers does not alter the values of either index). In this example, the two indices are positively correlated. However, the correlation becomes less predictable when the sum of integers varies across sets. For instance, in the two sets $j = \{14, 9\}$ and $k = \{8, 6\}$, the corresponding sums are 23 and 14, the Brillouin indices are 0.5918 and 0.5719 and the evenness indices are 0.9643 and 0.9835. An inverse correlation between the total count and the Brillouin index is possible. For instance, $w = \{7, 3\}$ (sum = 10) has a higher Brillouin index (0.4787) than $z = \{9, 3\}$ (sum = 12; index = 0.4494). Following on from these examples and to assist the reader in interpreting the polarity of changes, it is appropriate to view increases in the two indices as complementary but distinct facets of column 'complexification'. We produced graphic summaries of the index distributions across groups in the form of violin plots in vioplot[114] (Fig. 3) and probability density distributions in ggplot2 (ref. [115]) (Supplementary Data 1).

### Poisson regressions and phylogenetic analyses of variance

To test for differences in each of the CTL, TL, T and L counts between groups, we performed Poisson regressions of counts versus groups (Supplementary Table 1) using the glm.nb function in MASS[116] to build regression models. We used the Anova function in car to assess the degree and significance of parameter deviance from a null model. We used the nagelkerke function in rcompanion[77] to calculate pseudo-$R^2$ coefficients, measuring how well each of the Poisson regression models explains the data. In addition, we used emmeans[117] for conducting post hoc tests of significant pair-wise differences between estimated means of group-specific counts. Lastly, we evaluated significant differences among the mean values of each index across groups[78] (Supplementary Table 2) using the phylANOVA function in phytools[118].

### Independent contrasts and phylogenetic generalized least squares analyses

We used phylogenetic independent contrasts (brunch function in caper[83]) (Supplementary Table 3) to model the regression between TL counts and T:L ratios (Fig. 5a and Supplementary Fig. 1), as well as between each of the CTL and TL counts and their associated Brillouin and evenness indices (Supplementary Figs. 2–5). The 'brunch' regression model is suitable for predictors, such as counts, that take the form of ordered multinomial data. In addition, we used phylogenetic generalized least squares regression (pgls function in caper) (Supplementary Table 3) to correlate T:L ratios with each of the TL Brillouin and evenness indices (Supplementary Figs. 6 and 7). Comparative data objects for both brunch and pgls regressions used the phylogenetic variance–covariance matrix with the full array of individual branch lengths

contributing to the shared lengths between any two tips of the phylogeny. For each regression, we output diagnostic plots using the same functions. Given the heteroskedastic distributions of all indices, local variations in the distribution of values for the predictor and response values are more appropriately visualized through LOESS regression curves[119] in the bivariate scatterplots (note: log-transforming the variables, either each one separately or both together, does not remove the influence of heteroskedasticity).

### Continuous trait mapping

We used the contMap function in phytools to produce colour-coded plots of the complexity indices onto the phylogeny (Fig. 4 and Extended Data Fig. 1). Index values at nodes represent maximum likelihood estimates[80], while those along the branches are interpolated following methods in ref. [120].

### Descendant–ancestor tests

For each index, we calculated estimated values at internal nodes under maximum likelihood with the fastAnc function in phytools. We adapted protocols expounded in ref. [93] to correlate descendant–ancestor differences with ancestral node values across the entire tree (Fig. 6b,d,f,h and Extended Data Fig. 2b,d) and for each individual group (Supplementary Figs. 13–19) after correcting for the regression to the mean artefact. The regression to the mean is a statistical phenomenon whereby, due to imperfect correlations between variables (such as may be caused by sampling error and/or inadequately representative samples), the value of a variable tends to occur outside the 'norm' when first measured but it is likely to approach the population mean in subsequent measures[121]. As the regression to the mean is intrinsic to any imperfect correlation, we took it into account as integral to the regression procedure. The first step in this procedure involved testing for equality of variances associated with each of the sets of ancestor and descendant values. Depending upon outcome, one can use the Pearson's correlation coefficient '$r$' to adjust the descendant–ancestor differences when the equality of variances is not rejected or a modified correcting index when it is rejected. In the first scenario, let **X** and **Y** represent, respectively, vectors of ancestor and descendant values. The difference between **X** and **Y** can be modified as follows, to account for the regression to the mean artefact:

$$D = r \times (\mathbf{X} - \mathrm{mean}\,(\mathbf{X})) - (\mathbf{Y} - \mathrm{mean}\,(\mathbf{Y}))$$

In the second scenario, let s**X** and s**Y** be the standard deviations, respectively, of **X** and **Y** and var**X** and var**Y** their respective variances. Following ref. [122], we introduce a correcting factor, termed 'adj', given by the following formula:

$$\mathrm{adj} = (2 \times r \times s\mathbf{X} \times s\mathbf{Y}) / (\mathrm{var}\mathbf{X} + \mathrm{var}\mathbf{Y})$$

We then calculate adjusted differences as follows:

$$D1 = \mathrm{adj} \times (\mathbf{X} - \mathrm{mean}\,(\mathbf{X})) - (\mathbf{Y} - \mathrm{mean}\,(\mathbf{Y}))$$

Note: the sign of these differences was subsequently reversed to ensure that negative and positive differences (descendant minus ancestor value) represent, respectively, decreases and increases. We tested for significant differences between increases and decreases (positive and negative $D1$ values, respectively) in three ways using base R functions. First, we established whether the number of increases and the number of decreases deviated from a 1:1 null proportional distribution with a binomial (sign test) of equal proportions. Second, we tested whether the mean of all $D1$ values differed from zero using a two-tailed, one-sample Wilcoxon signed rank test. Third, we compared the absolute mean of all negative $D1$s with the mean of all positive $D1$s through an unpaired, two-tailed, two-sample Wilcoxon rank sum test. Following

these preliminary characterizations of the distributions of $D1$ values and to establish whether large or small ancestral values tend to be associated chiefly with large or small descendant–ancestor differences, we carried out robust linear regressions in MASS (Supplementary Table 4). The significance of the slope and intercept of all regression models was established with a heteroskedasticity-robust Wald $F$-test using sfsmisc. Robust linear regressions were further applied to individual groups. To provide direct tests of the overall direction of index variation over time, we undertook robust linear regressions of ancestor values against node ages in MASS for the entire taxon set (Fig. 6a,c,e,g and Extended Data Fig. 2a,c) and for individual groups (Supplementary Figs. 8–12). The node ages were obtained in paleotree[123] (dateNodes function).

### Subclade tests

To establish whether trends conform to passive or driven processes, we applied a subclade test of skewness[45]. The test operates by partitioning the total skewness of a trait in a parent group into three main components: skewness between subclades (SCB); skewness within subclades (SCW); and heteroskedasticity-related skewness (SCH; skewness caused by heterogeneity in trait variance in subclades) (Supplementary Figs. 20–24, Extended Data Fig. 3 and Supplementary Table 5). The test quantifies the combined effects of passive and driven processes in terms of the proportional contributions of SCB, SCW and SCH to the total skewness. Such contributions are based upon the normal versus non-normal distributions of mean subclade values relative to the mean value of the parent group and upon the degree of skewness in the subclades (that is, small versus large standard deviations of subclade values within the right tail of the total group). The prevalence of either SCB or SCH suggests passive trends, while the prevalence of SCW suggests driven trends. High proportions of SCW suggest that the overall skewness pattern is replicated in several constituent subclades. This is itself indicative of a driven trend, as a tendency towards higher or lower values will skew not only the ensemble distribution but also the distributions of most or all subclades[45,84,88–90]. As an example, consider a parent group with a right-skewed trait (for example, complexity or size). Furthermore, suppose that each constituent subclade shows symmetrically distributed values around its own mean and that the means of the subclade distributions are right-skewed around the parent group's mean. In this scenario, the right-skewed distribution of the parent group results from a passive trend, as SCB prevails. Now assume that the distribution of each subclade is also right-skewed. In this scenario, SCW prevails, pointing to a driven trend. This would also be the case if the means of the subclades were symmetrically distributed around the parent group's mean. In a final scenario, suppose that each subclade has symmetrically distributed values and, further, that the variance increases in subclades at the right tail end of the parent group's distribution (that is, those subclades exhibit a greater spread of values around their own means). In this case, the right-skewed distribution of the parent group is caused by heteroskedasticity (SCH) and, as in the case of SCB, it indicates a passive process[45]. The test code builds probability density functions for the values of the parent group and those of its subclades and outputs a list of the percentage contributions of SCB, SCH and SCW to the skewness of the parent group. For each index, the total skewness was calculated in e1071. As analyses of skewness are predicated on right-skewed distributions[45,84] and because most indices are negatively skewed (except for unstandardized T:L), we transformed those indices by taking their negative logarithms[89] before subjecting them to the test. Subclade tests on selected major groups followed identical protocols.

### Analyses of rates

We used reversible-jump Markov Chain Monte Carlo methods in geiger[69] to detect shifts in rates of complexity changes[70] (Extended Data Figs. 4 and 5). For each tested index, we ran the rjmcmc.bm function for $5 \times 10^7$ generations under a relaxed-clock Brownian motion model of evolution, sampling every 500 generations. Under relaxed-clock Brownian motion, changes are permitted following two modalities: (1) unusually high or low rates occurring on single branches, resulting in a shift in global optimum along those branches and (2) shifts in rate of evolution inherited by all descendants of a given branch. After each run, we checked chain convergence towards equilibrium in Tracer[124] and visualized the posterior Bayesian rate estimates (median values) and rate shifts on the tree using plotting functions in geiger. Branches showing median rates that are higher or lower than the background rates are colour-coded with gradients of colour intensity (maroon for higher rates and steelblue for lower), where greater intensity signifies greater rates of change. The posterior probabilities of rate shifts are indicated by solid circles, the size of which is proportional to their posterior probabilities and with colour coding identical to that used for the branch rates (maroon, upturn shift; steelblue, downturn shift).

### Reporting summary

Further information on research design is available in the Nature Portfolio Reporting Summary linked to this article.

### Data availability

The data that support the findings of this study are available in Figshare and accessible at https://doi.org/10.6084/m9.figshare.21622284. Supplementary Tables 1–5 include outputs from the following analyses: Poisson regressions of counts versus groups; phylogenetic analyses of variance for the complexity indices; phylogenetically corrected correlations between various categories of vertebral counts and complexity indices; robust linear regressions between ancestral complexity values and descendant–ancestor differences, as well as between node ages and ancestral values; and subclade tests. Supplementary Data 1 lists taxa, their presacral counts and their complexity indices. It also reports univariate statistics and histogram distributions for the thoracic and lumbar counts alongside probability density distributions for various complexity indices. The literature sources on vertebral formulae are listed in Supplementary Data 2. The time-scaled phylogeny is available in Supplementary Data 3 as an object of class 'phylo'. R code is reproduced in Supplementary Data 4 and is accompanied by templates for running analyses on individual data files extracted from Supplementary Data 1. Such data files are combined as separate tabs within individual spreadsheets and are available as Source Data for Figs. 3–6 and for Extended Data Figs. 1–5. These Source Data can be redeployed for building Supplementary Figs 1–24.

### Code availability

All results and graphics are reproducible using the protocols detailed in the Methods and the examples accompanying the relevant functions of various packages. R code is deposited in Figshare and accessible at https://doi.org/10.6084/m9.figshare.21622284.

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

## Acknowledgements

We are grateful to the John Templeton Foundation (grant no. 61408 to M.W. and M.R.), BBSRC (grant nos BB/K015702/1 and BB/K006754/1 to M.W.) and NERC (studentship 2276912 to A.B. and M.W.) for supporting this research. For the purpose of Open Access, the authors have applied a CC BY public copyright licence to any Author Accepted Manuscript version arising from this submission. We thank R. Benson (University of Oxford) for providing insightful remarks on an earlier draft of this work and for allowing us to use the three-dimensional reconstructions of the vertebral columns in Fig. 1. Such reconstructions were skilfully rendered from computerized tomography scans by E. Griffiths (University of Oxford). S. Wang (Swarthmore College) kindly supplied a copy of his skewness test. We extend our gratitude to L. Barber for her stalwart efforts in locating important literature sources and O. Wills for her beautiful rendition of Fig. 1. We also thank T. Michael Keesey for obtaining the icons in Fig. 2 from PhyloPic.

## Author contributions

M.R., M.W. and Y.L. conceived the project. E.G., Y.L. and M.R. collected the data. A.B., E.G., Y.L., J.O., M.R. and M.W. conceptualized analytical and statistical protocols. A.B., J.O. and M.R. supplied codes. A.B., E.G., Y.L., J.O. and M.R. analysed the data. M.R. tabulated results, produced figures and tables and wrote an initial draft of the manuscript. All authors contributed equally to the final draft of the work.

## Competing interests

The authors declare no competing interests.

## Additional information

**Extended data** is available for this paper at https://doi.org/10.1038/s41559-023-01982-5.

**Correspondence and requests for materials** should be addressed to Matthew Wills or Marcello Ruta.

[1]Milner Centre for Evolution, Department of Biology and Biochemistry, University of Bath, Bath, UK. [2]Nanjing Institute of Geology and Palaeontology, CAS, Nanjing, China. [3]Joseph Banks Laboratories, Department of Life Sciences, University of Lincoln, Lincoln, UK. ✉e-mail: bssmaw@bath.ac.uk; mruta@lincoln.ac.uk

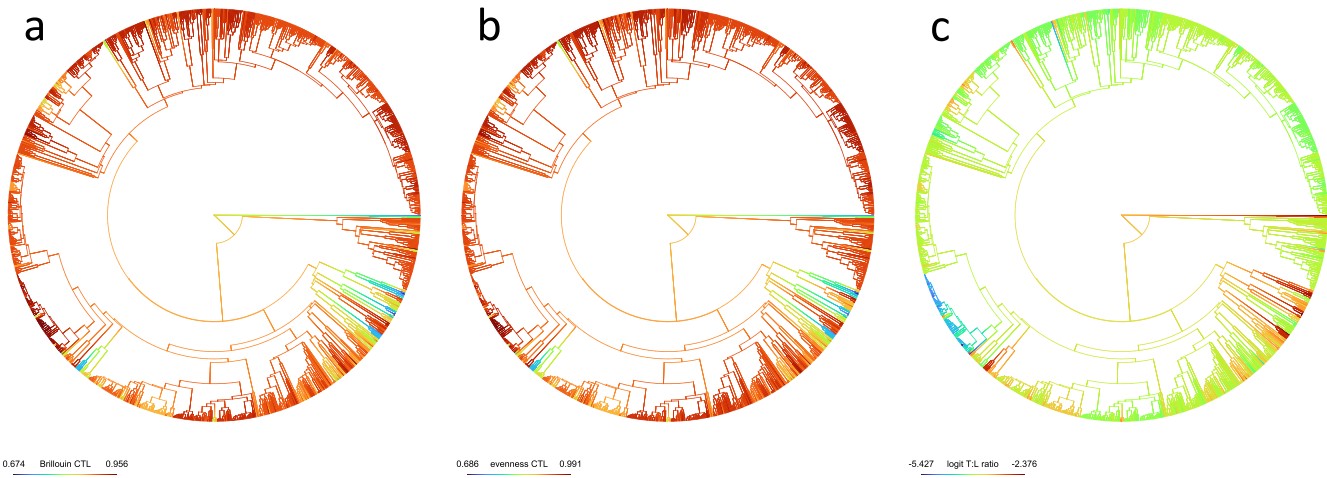

**Extended Data Fig. 1 | Continuous mapping of complexity indices across the phylogeny.** The values of three complexity indices are mapped onto the phylogeny using colour gradients. Index values at the internal nodes are estimated through maximum likelihood, and those along the branches are interpolated between the nodal estimates. For each index, the colour scales range from its minimum to its maximum value. The lengths of the scale bars are in millions of years. **a**, Presacral Brillouin index. **b**, Presacral evenness index. **c**, Logit thoracic:lumbar ratio.

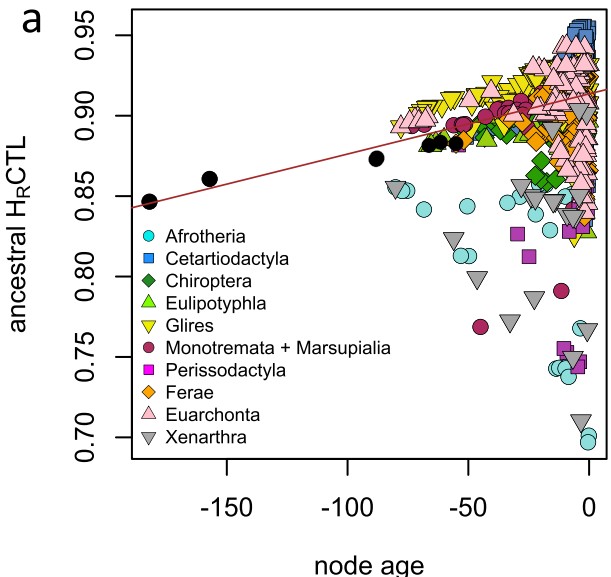

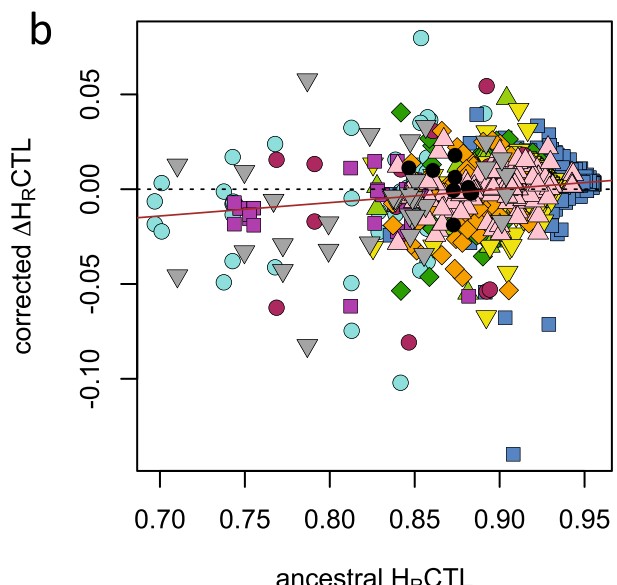

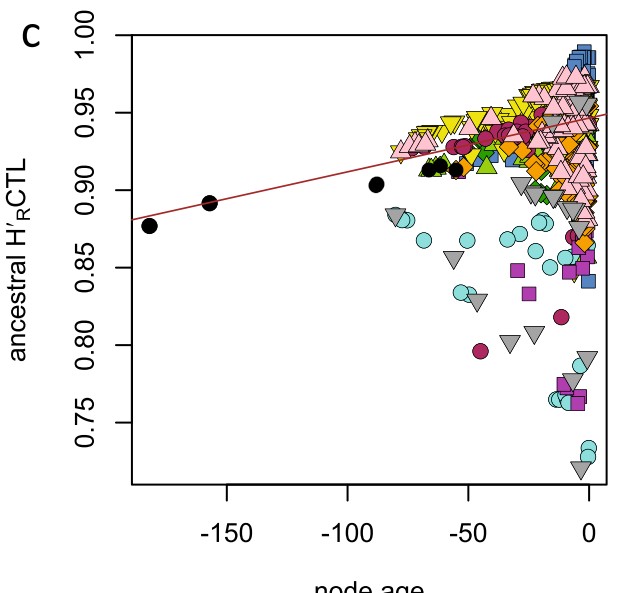

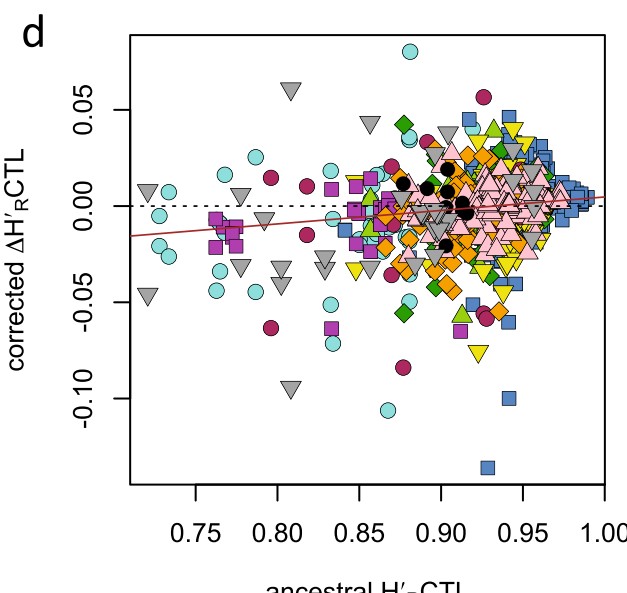

**Extended Data Fig. 2 | Bivariate plots of node estimates of complexity indices vs. node ages and descendant–ancestor differences vs. node estimates.** Robust linear regressions between maximum likelihood node estimates of two complexity indices and node ages (**a**, **c**), and between descendant–ancestor differences (corrected for the regression to the mean) and node estimates (**b**, **d**). The brown solid lines are regression lines. The black dashed horizontal lines separate positive (increases) from negative (decreases) descendant–ancestor differences. Node ages are in millions of years, with time decreasing in the positive direction of the axis (that is, closer to the present). Mammal groups are identified by distinct colours and symbols. Black circles denote the deepest nodes of the phylogeny, corresponding to the separation between major mammal cohorts. **a-b**, Presacral Brillouin index. **c-d**, Presacral evenness index.

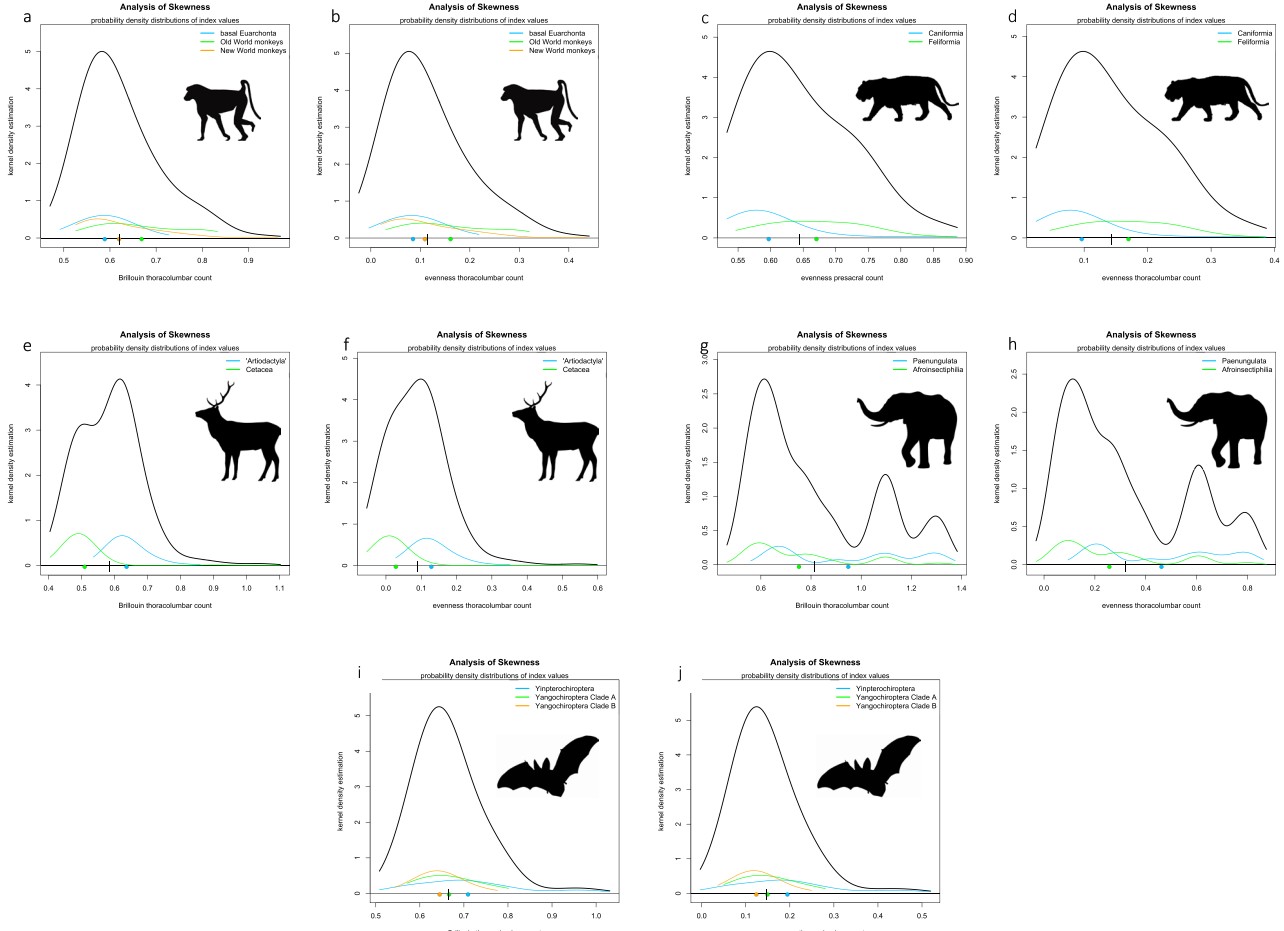

**Extended Data Fig. 3 | Subclade tests in selected mammal groups.** Results of skewness partitioning tests applied to the thoracolumbar Brillouin and evenness indices in various mammal groups. In each plot, the probability density distributions of the index values for each subclade are shown by colour-coded thin lines, whereas the index distribution for the entire group is shown by a thick black curve. The mean values of the individual subclades are represented by colour-coded circles, whereas the mean value of the entire distribution is marked by a black vertical bar. **a-b**, Euarchonta. **c-d**, Carnivora. **e-f**, Cetartiodactyla. **g-h**, Afrotheria. **i-j**, Chiroptera. Image credits for mammal silhouettes are as in Fig. 1 of the main text.

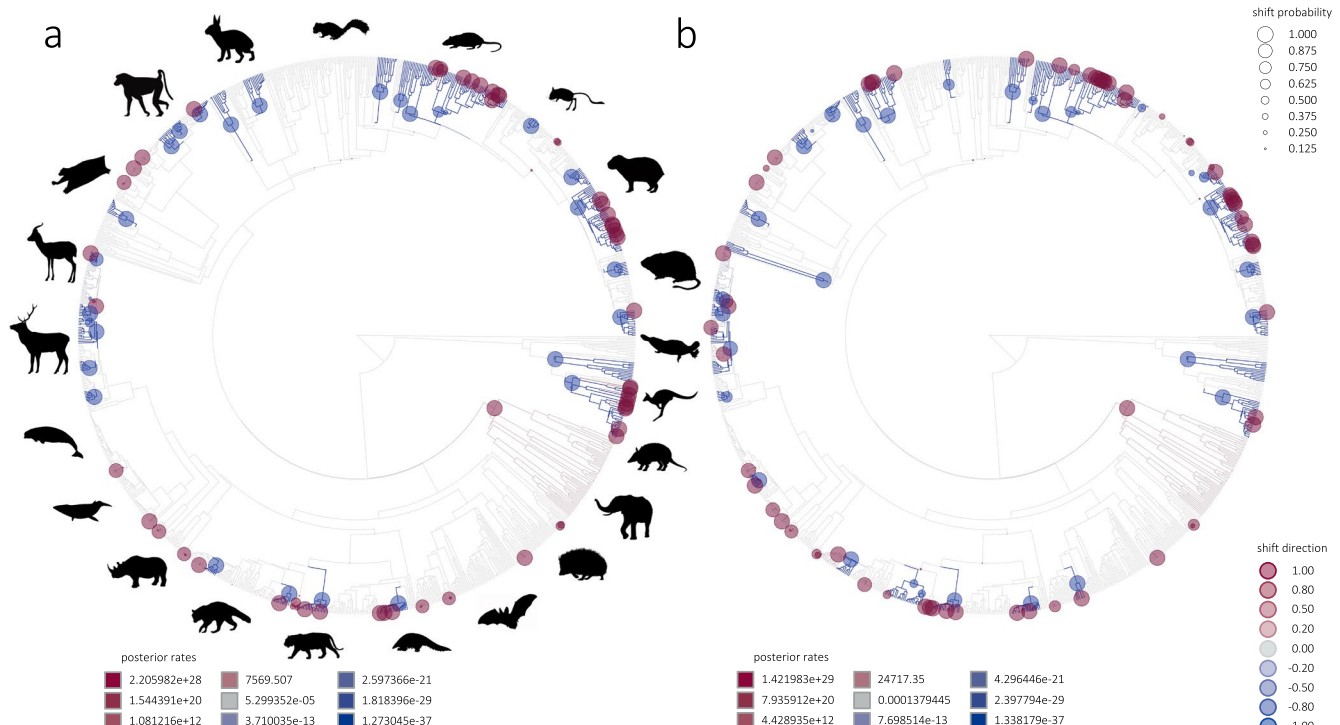

**Extended Data Fig. 4 | Shifts in rates of complexity change for the thoracolumbar region.** Colour-coded evolutionary rates and shifts mapped onto the phylogeny. Grey branches exhibit background rates. Maroon and steelblue branches exhibit rates that are, respectively, higher and lower than the background rates. Colour intensity is proportional to the rate values, with darker tones indicating a greater difference between background and non-background rates. The circles mark the locations of rate shifts. Circle sizes are drawn in proportion to the posterior Bayesian probability of shifts. Circle colours represent shift magnitude, with darker maroon (respectively, steelblue) tone indicating a shift of greater magnitude towards a rate increase (respectively, decrease) relative to the rates of adjacent branches. **a**, Thoracolumbar Brillouin index. **b**, Thoracolumbar evenness index. Image credits for mammal silhouettes are as in Fig. 1 of the main text.

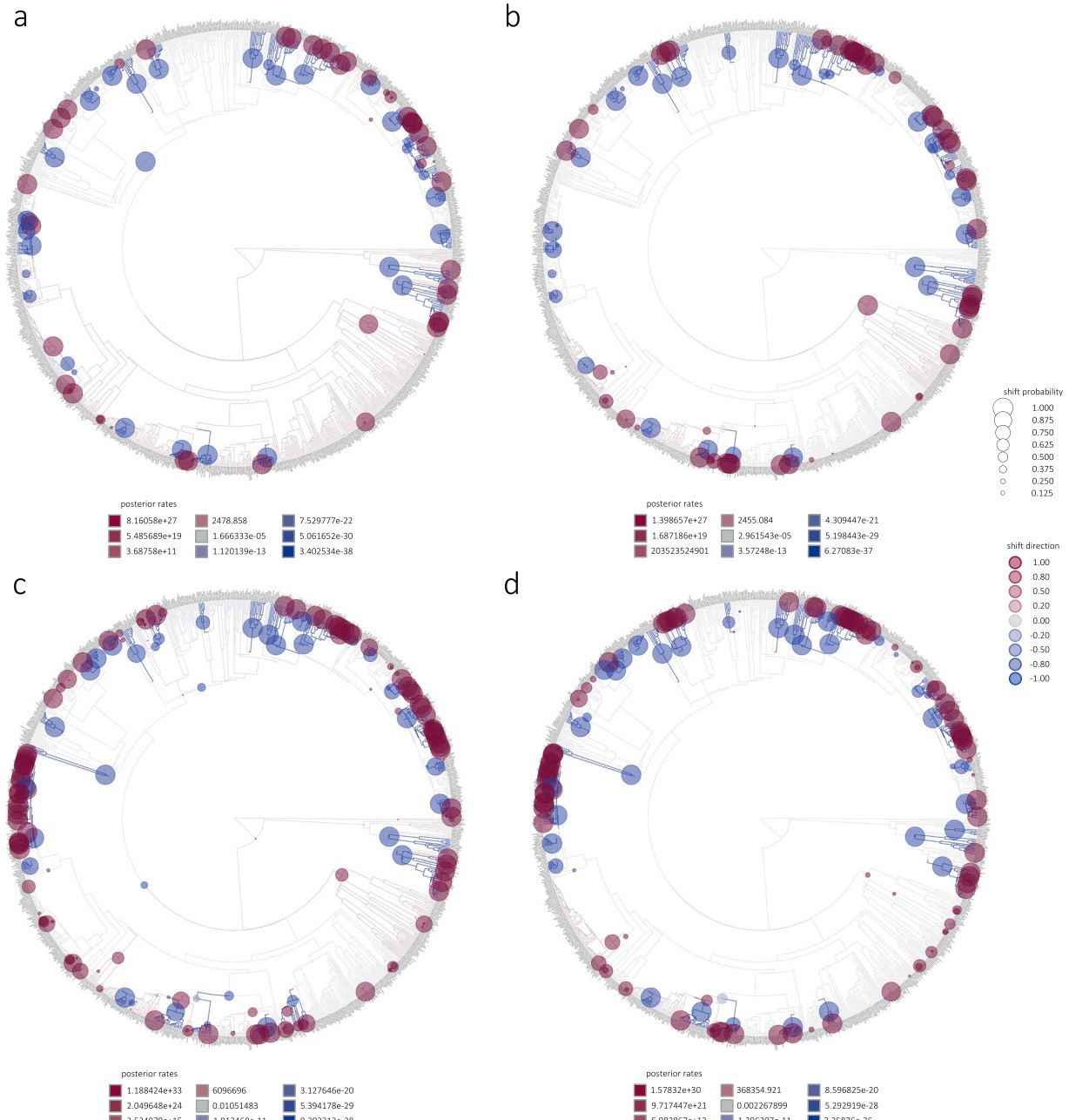

**Extended Data Fig. 5 | Shifts in rates of complexity change for the presacral region and thoracolumbar domains.** Colour-coded evolutionary rates and shifts mapped onto the phylogeny. For explanations of colours and symbols, see caption of Extended Data Fig. 4. **a**, Presacral Brillouin index. **b**, Presacral evenness index. **c**, Unstandardized thoracic:lumbar ratio. **d**, Logit-transformed thoracic:lumbar ratio.

# Reporting Summary

## Statistics

For all statistical analyses, confirm that the following items are present in the figure legend, table legend, main text, or Methods section.

| n/a | Confirmed | |
|---|---|---|
| ☐ | ☒ | The exact sample size (*n*) for each experimental group/condition, given as a discrete number and unit of measurement |
| ☐ | ☒ | A statement on whether measurements were taken from distinct samples or whether the same sample was measured repeatedly |
| ☐ | ☒ | The statistical test(s) used AND whether they are one- or two-sided *Only common tests should be described solely by name; describe more complex techniques in the Methods section.* |
| ☐ | ☒ | A description of all covariates tested |
| ☐ | ☒ | A description of any assumptions or corrections, such as tests of normality and adjustment for multiple comparisons |
| ☐ | ☒ | A full description of the statistical parameters including central tendency (e.g. means) or other basic estimates (e.g. regression coefficient) AND variation (e.g. standard deviation) or associated estimates of uncertainty (e.g. confidence intervals) |
| ☐ | ☒ | For null hypothesis testing, the test statistic (e.g. $F$, $t$, $r$) with confidence intervals, effect sizes, degrees of freedom and $P$ value noted *Give P values as exact values whenever suitable.* |
| ☐ | ☒ | For Bayesian analysis, information on the choice of priors and Markov chain Monte Carlo settings |
| ☒ | ☐ | For hierarchical and complex designs, identification of the appropriate level for tests and full reporting of outcomes |
| ☒ | ☐ | Estimates of effect sizes (e.g. Cohen's *d*, Pearson's *r*), indicating how they were calculated |

*Our web collection on statistics for biologists contains articles on many of the points above.*

## Software and code

Policy information about availability of computer code

| Data collection | No specialist or commercial software was used |
|---|---|
| Data analysis | R code for running analyses is available as part of supplementary information (Supplementary Data 4) |

For manuscripts utilizing custom algorithms or software that are central to the research but not yet described in published literature, software must be made available to editors and reviewers. We strongly encourage code deposition in a community repository (e.g. GitHub). See the Nature Portfolio guidelines for submitting code & software for further information.

## Data

Policy information about availability of data

All manuscripts must include a data availability statement. This statement should provide the following information, where applicable:
- Accession codes, unique identifiers, or web links for publicly available datasets
- A description of any restrictions on data availability
- For clinical datasets or third party data, please ensure that the statement adheres to our policy

The data that support the findings of this study are available in Figshare and accessible at: https://doi.org/10.6084/m9.figshare.21622284
Supplementary Tables 1–5 include outputs from the following analyses: Poisson regressions of counts vs. groups; phylogenetic analyses of variance for the complexity indices; phylogenetically corrected correlations between various categories of vertebral counts and complexity indices; robust linear regressions between ancestral complexity values and descendant-ancestor differences, as well as between node ages and ancestral values; subclade tests. Supplementary Data

1 lists taxa, their presacral counts, and their complexity indices. It also reports univariate statistics and histogram distributions for the thoracic and lumbar counts alongside probability density distributions for various complexity indices. The literature sources on vertebral formulae are listed in Supplementary Data 2. The time-scaled phylogeny is available in Supplementary Data 3 as an object of class 'phylo'. R code is reproduced in Supplementary Data 4 and is accompanied by templates for running analyses on individual data files extracted from Supplementary Data 1. Such data files are combined as separate tabs within individual spreadsheetsand are available as Source Data for main Figures 3–6 and for Extended Data Figures 1–5.

# Human research participants

Policy information about studies involving human research participants and Sex and Gender in Research.

| | |
|---|---|
| Reporting on sex and gender | *Use the terms sex (biological attribute) and gender (shaped by social and cultural circumstances) carefully in order to avoid confusing both terms. Indicate if findings apply to only one sex or gender; describe whether sex and gender were considered in study design whether sex and/or gender was determined based on self-reporting or assigned and methods used. Provide in the source data disaggregated sex and gender data where this information has been collected, and consent has been obtained for sharing of individual-level data; provide overall numbers in this Reporting Summary. Please state if this information has not been collected. Report sex- and gender-based analyses where performed, justify reasons for lack of sex- and gender-based analysis.* |
| Population characteristics | *Describe the covariate-relevant population characteristics of the human research participants (e.g. age, genotypic information, past and current diagnosis and treatment categories). If you filled out the behavioural & social sciences study design questions and have nothing to add here, write "See above."* |
| Recruitment | *Describe how participants were recruited. Outline any potential self-selection bias or other biases that may be present and how these are likely to impact results.* |
| Ethics oversight | *Identify the organization(s) that approved the study protocol.* |

Note that full information on the approval of the study protocol must also be provided in the manuscript.

# Field-specific reporting

Please select the one below that is the best fit for your research. If you are not sure, read the appropriate sections before making your selection.

☐ Life sciences     ☐ Behavioural & social sciences     ☒ Ecological, evolutionary & environmental sciences

For a reference copy of the document with all sections, see nature.com/documents/nr-reporting-summary-flat.pdf

# Ecological, evolutionary & environmental sciences study design

All studies must disclose on these points even when the disclosure is negative.

| | |
|---|---|
| Study description | We analyze the evolution of complexity in serial structures, using the mammal vertebral column as a case-study. We derive indices of numerical abundance and proportional distributions of vertebral types and subject them to comparative phylogenetic methods, testing for occurrences of rate heterogeneities and evolutionary trends in complexity |
| Research sample | Vertebral data on 1136 mammal species |
| Sampling strategy | Taxa covered in this study belong to all major groups of extant mammals |
| Data collection | Online data collection of vertebral formulae from a representative sample of extant mammal species |
| Timing and spatial scale | Online data collection started approximately two years ago and was completed near the end of 2021 |
| Data exclusions | None of the collected data were excluded from the analyses |
| Reproducibility | All results can be reproduced with codes made available as part of supplementary information (Supplementary Data 4) |
| Randomization | The nature of our study does not involve random assignments of samples to groups. Species are assigned to major mammal groups following current taxonomy |
| Blinding | All members of the team re-ran analyses to confirm results |

Did the study involve field work?     ☐ Yes     ☒ No

# Reporting for specific materials, systems and methods

We require information from authors about some types of materials, experimental systems and methods used in many studies. Here, indicate whether each material, system or method listed is relevant to your study. If you are not sure if a list item applies to your research, read the appropriate section before selecting a response.

| Materials & experimental systems | Methods |
|---|---|

**Materials & experimental systems**

| n/a | Involved in the study |
|---|---|
| ☒ | Antibodies |
| ☒ | Eukaryotic cell lines |
| ☒ | Palaeontology and archaeology |
| ☒ | Animals and other organisms |
| ☒ | Clinical data |
| ☒ | Dual use research of concern |

**Methods**

| n/a | Involved in the study |
|---|---|
| ☒ | ChIP-seq |
| ☒ | Flow cytometry |
| ☒ | MRI-based neuroimaging |

