## [Peer Review File · Nature Ecology & Evolution]

Peer Review Information

Journal: Nature Ecology & Evolution

Manuscript Title: Divergent vertebral formulae shape the evolution of anatomical complexity in mammals

Corresponding author name(s): Marcello Ruta

Editorial Notes:

Reviewer Comments & Decisions:

Decision Letter, initial version:

14th March 2022

Dear Dr Ruta,

Your manuscript entitled "Divergent vertebral formulae shape the evolution of complexity in mammals" has now been seen by three reviewers, whose comments are attached. The reviewers have raised a number of concerns which will need to be addressed before we can offer publication in Nature Ecology & Evolution. We will therefore need to see your responses to the criticisms raised and to some editorial concerns, along with a revised manuscript, before we can reach a final decision regarding publication.

We therefore invite you to revise your manuscript taking into account all reviewer and editor comments. Please highlight all changes in the manuscript text file [OPTIONAL: in Microsoft Word format].

* If you have not done so already please begin to revise your manuscript so that it conforms to our Article format instructions at <http://www.nature.com/natecolevol/info/final-submission>. Refer also to any guidelines provided in this letter.

2[REDACTED]

Nature Ecology & Evolution is committed to improving transparency in authorship. As part of our efforts in this direction, we are now requesting that all authors identified as 'corresponding author' on published papers create and link their Open Researcher and Contributor Identifier (ORCID) with their account on the Manuscript Tracking System (MTS), prior to acceptance. ORCID helps the scientific community achieve unambiguous attribution of all scholarly contributions. You can create and link your ORCID from the home page of the MTS by clicking on 'Modify my Springer Nature account'. For more information please visit <http://www.springernature.com/orcid>.

[REDACTED]

Reviewer expertise:

All reviewers have expertise in mammalian vertebral evolution

Reviewers' comments:

Reviewer #1 (Remarks to the Author):

This manuscript seeks to address macroevolutionary patterns of complexity in the vertebral column of mammals. The authors test the long-held idea that complexity increases via a driven process (as an intrinsic evolutionary law) as opposed to via random or passive processes. This is an interesting topic that has received much attention in terms of theory, but few tests with empirical data. Therefore, I think this study makes a nice contribution, with a nice large sample, and very appropriate methods. The paper is very well written, and generally requires little improvement. The methods are appropriate, thorough, and well explained. Below I provide some comments that I think could improve the paper, but these are mostly to improve readability. I look forward to seeing this published.

2Introduction –

- It would really help readers who are not familiar with vertebrae to include an introductory figure with a vertebral column indicating how your metrics were calculated (indices, ratios). I think the paper will appeal to a broad audience, so they might not be familiar with the anatomy. Plus, given the paper is all about vertebrae, it would be nice to see at least one vertebral column! Is low evenness an indicator of higher complexity – the polarity is not obvious to me with this?
- At the end of the introduction, it would be helpful to clearly state the hypotheses you are testing. Without this, due to the number of analyses, it can seem like a bit of a laundry list of analyses without being clear how each relates to a specific question or hypothesis.

Results/discussion

- P6 – you mention that many distributions are bimodal. Why do you think this is? Does this suggest a need for more specific subgroup analyses (ie. Capturing two distinct patterns)?
- It is not clear to me the benefit of calculating the indices on both the presacral and TL regions. Cervical counts are fixed at 7 in mammals (except for rare exceptions), so we would not expect this region to contribute to variation. They seem to show very similar results. It might make the paper more streamlined if you only present one of these in the main text, and move the other to supplement.
- P7, l156 – “the presacral column features a substantial degree of numerical heterogeneity and evenness”. Again, I think this is very hard for the reader to imagine. Can you describe an example to illustrate or put it in the figure above? E.g., a column in which the proportion of vertebrae in each region is similar. Including these more descriptive statements will really help with interpretation.
- P8 – why might cetaceans vary T and L counts in similar proportions? Think its worth mentioning that vertebral morphology is far more homogeneous in cetacea than terrestrial mammals, suggesting less functional differentiation, and perhaps an absence of distinct selection pressures that would cause the ratio to vary.
- P9, l210 – Any more interpretation around this pattern? Does this suggest that vertebral columns tend to elongate via the disproportionate addition of thoracics, particularly in shorter columns.
- P13 – Is it correct in assuming that the contribution of SCW reflects the relative importance of the driven trends i.e., complexity increases are approximately 2/3 passive and 1/3 driven?

Conclusion

- P15 – “We uncover modest but non negligible evidence of driven trends towards increasing column regionalization and evenness.” – I don’t understand what you mean by regionalization here, surely regionalization would be the opposite of evenness? Do you mean a driven trend in T/L, in which case that would seem to be an increase in relative thoracic length ie preferential addition of thoracics. Would increasing evenness mean decreasing complexity?

Figures

- Fig 2 – add key or heading to each figure. It has been shown that this default rainbow color ramp is can be misleading and isn’t color blind friendly. Consider using alternatives in r e.g., vidiris. There are a lot of similar contmaps here – can you move any to supplement e.g. TL vs presacral?
- Fig 4 – again – do we need both presacral and TL in main text (they are pretty much identical)?
- Fig 5 – remove species names as they are too small to read. Instead adding silouhettes and key node names as in figure one would be more helpful.

Katrina Jones

Reviewer #2 (Remarks to the Author):

These researchers are interested in evolutionary complexity and chose regional numbers of vertebrae to test the hypothesis of conservation of presacral numbers of vertebrae in mammals. As such, they tackle major issues in evolutionary biology and find interesting results, namely that presacral numbers of vertebrae, and thoracolumbar numbers specifically, are not conserved interspecifically and across clades in mammals as they are broadly thought to be. They do so using phylogenetic analyses and various indices that quantify count data. I am favorable to their approach and the methods employed seem appropriate and creative. The breadth of the study is impressive, as are the beautiful but nearly unwieldy figures. I see this as a welcome contribution but have one major question/concern/request regarding data, then a series of minor edits and suggestions that are listed below that, indicated by Line number (L).

Methods: The authors state that they scour google scholar for vertebral counts and synopses from the journal Mammalian Species (MS). Are these two separate processes or one in the same (i.e., are the authors getting these data directly from Mammalian Species and nowhere else)? Generally, I like the idea of sticking a primary source (even if it's a journal), and I wonder if each individual reference paper needs to be cited. If these are separate processes and data are coming from any manuscript with the terms searched for, I think these certainly need to be credited and cited somewhere. I do not see a list of these or separate bibliography for them anywhere in the supplementary materials. This is why I assume the primary source is MS. But that could be incorrect and it could be that the authors still should cite the individual papers the data come from even if from MS.

Methods: It is also stated that one author collected data using first-hand gathering at two museums. Looking over the supplementary materials, I cannot tell which species are represented by published records and which were gathered in novel data collection at museums. It would be important to know what percentage was gathered using each process, if there was overlap (e.g., a record was found for *Canis lupus* and the MR also counted vertebrae in a specimen), and, ideally, the species would be coded according to where the data came from (e.g., published record, museum data, mixed, or the like). It would also be extremely valuable (although perhaps not desired or even practical here) to publish the raw data; that is, specimen numbers and their associated vertebra counts (see Sanchez-Villagra et al., 2007).

L110: Add comma between "vertebrae" and n1.

L235: Should Evenness be capitalized or not? It is capitalized here and in some other places but not in others (e.g., Ls319, 519).

L328: Old World should be capitalized and does not require a hyphen (although it can be used).

L341: Something is missing from the part of the sentence with "we remark."

L393-394: The refer to both colugos and tree shrews as gliding taxa. Colugos are gliders, but tree shrews are not; rather, they are scansorial quadrupeds. Additionally, they do not have longer thoracic columns and shorter lumbar columns (with the exception of *Ptilocercus*, which demonstrates this pattern only slightly).

L434: Wording is mixed up and/or something is missing from this sentence around "may seldom alter significantly way the."

Reviewer #3 (Remarks to the Author):

In this work, Li et al. aim to examine the evolution of vertebral count in mammals by compiling a large species sample size (>1000) and using various measures of complexity: two from information theory (often applied to species diversity) including Brillouin index ("numerical diversity") and the related Evenness index ("numerical distribution"), while the other two related to thoracic+lumbar counts and T/L ratios. These complexity metrics are then explored using (1) maximum-likelihood ancestral state reconstruction, (2) count-ratio correlations, (3) evolution of vertebral count complexity values using ancestor-descendant analyses over time, (4) passive vs. driven evolution via a subclade (skewness) test, and (5) evolutionary rate shifts across the mammal phylogeny.

The primary conclusions from this study are that the number of presacral vertebrae across mammals is variable, the mean complexity values across major clades do not significantly differ, count is positively related to increases in diversity and evenness but negatively related to T/L ratios. Further, ancestor-descendant relationship shows an overall trend to increasing complexity but that this trend is mainly passive ("with a non-negligible driven trend"). Finally, there are shifts towards both increasing and decreasing complexity across the mammalian tree.

Overall, this study was jam packed with data and took an interesting spin on analyzing vertebral count evolution making the foundation work novel. I appreciate the hard work of collecting and analyzing such a large dataset and commend the authors for presenting such a large phylogenetic study using a large suite of methods. However, the study, for me, falls a bit flat as there is little interpretation of the results presented – what does this all mean for mammal (vertebral) evolution or complexity more broadly? The Results & Discussion section is primarily just a summary of the results with little interpretation and the Conclusions is more of a broad literature review as well as discussion on how the next study will include fossils with some preliminary observations. While the Conclusions section does bring up prior work on aspects of ecology, development, biomechanics etc. that may influence vertebral morphology and count, these aspects are not tied back to the current study and the data presented. As currently presented, it is hard to see what the data mean in terms of mammalian (vertebral) evolution or the evolution of organismal complexity – what is underlying the count variability presented and the patterns recovered? In sum, the data are great, but the execution of the manuscript has not quite reached the level necessary to provide insight into the process of evolution.

- General comment: A major conclusion of this study is that it overturns long held ideas that mammals have fixed vertebral counts. However, this study is not the first to suggest variable counts (outside the cervical region) in mammals. Rodents have often been considered the least variable with respect to vertebral counts (that 19 TL mentioned), but various other groups have been shown to be much more variable in count including Afrotheres, Xenarthrans, Cetaceans, and some Primates. These groups also pop out as variable in the present analysis. There have also been a number of studies correlating variable counts across mammalian phylogenies with different locomotor modes (like suspensory locomotion or cursorial locomotion – both recent studies and referenced by the authors). Because of this, it feels a bit too strong to state emphatically that this study overturns long held ideas – but it does support recent thought and data on the column being much more variable than previously acknowledged.

5- Introduction: When I read the Introduction, I couldn't help but be reminded of the recent study by Jones and colleagues (2019, NatComms) on the evolution of vertebral complexity in mammals using information theory. Their study was a spinoff of McShea (1993, Evolution) that aimed to use information theory to examine mammal vertebral complexity in select ancestor-descendant crown mammal pairs; he found no evidence for a driven trend. The Jones paper specifically modeled various hypotheses for the evolution of complexity across fossil and extant synapsids as a whole using evolutionary modeling techniques and found evidence for stepwise evolution driven by the release of a function constraint and selection for increased metabolic rate – debunking passive or long-term trends of complexity during the early stages of synapsid and crown mammal evolution. While these two papers are cited in the present work, there is no specific discussion of them in the Introduction (and very little in the Discussion), considering their direct relevance – even if they used morphological complexity instead of count complexity.

- Mammal tree. This section does not seem like result or discussion. Perhaps some of this can be worked into the end of the Introduction.

- Clades – general comment: Some of the clades here are very large and contain many subclades with diverse ecological and locomotor habits. It would be nice to dig into these subclades more to better understand what might be driving some of the patterns recovered. Related to this, Supplementary Dataset 1 does not separate major clades by subclade so it is challenging to trace which groups might be driving any specific patterns.

- Complexity shows marked within-group variation. I found this section very hard to follow. One of the major problems is Figure 2 in that it lacks legends and does not give any indication of what the colors mean. In some instances, statements about groups being “high” and “intermediate” etc all looked the same. The color palette is just too hard to follow (note, Red is often used for high or fast and Blue for low or slow). Also, I needed to go back and forth to Figure 1 to figure out what node or clade I was looking at. There needs to be some guideposts if you are going to draw out specific group comparisons. This section also (and Figure 2 caption) referred to “heterogeneity” for the Brillouin index instead of “numerical diversity” as presented in the Introduction. The violine plots of within and between group variation feels like it should be in the main paper as it provides an important summary of the data collected (why are monotremes and marsupials lumped together here when they are vastly different clades?) and I kept going back and forth to confirm observations. Finally, while indices vary between clades they all have similar means – this is one criteria often used for passive evolution (increases and decreases equally likely).

- Vertebral numbers differ significantly between groups. Post-hoc tests recover differences but how many pair-wise tests were done? Put another way, what percentage of your pair-wise tests were different and were there any specific groups governing these trends? I notice that Afrotheria and Cerartiodactyla – both groups with extensive ecological variation – seem to dominate. With respect to count and ratio correlations, IC of T/L and T+L only explains about 2% of the variance in the data and the data are very scattered. Should you draw a line through this? To me, the only tight relationship here is between T/L and lumbar count, the other two are variable and driven by certain clades.

- Complexity trends are underpinned by passive and driven processes. The first part of this section

generally repeats the methods – can it be summarized more succinctly? What are your interpretations for the ancestor-descendant plots?. Is complexity increasing over time or is it random? The general results are stated but these are not interpreted in any meaningful way. In terms of skewness, it seems that the overriding trend is passive with some “non-negligible” driven force (the table of results has some odd values for Logit T/L). If there is a driven trend, then what is creating the “hard bound” if anything? If it is passive, why? This test is for the whole dataset but are you able to do similar tests with different ancestral- descendant populations? For instance, do some groups evolve passively while others via a driven trend? Perhaps this might allow you to interpret your data from physiological, biomechanical, or developmental perspective?

- Rate shifts: The results show widespread rate shifts across the tree – both increases and decreases. In McShea (1993) he states “if a forcing mechanism has operated, the expectation is that complexity would have increased in evolutionary lineages more frequently than it decreased.” What do you see in your data? Is there a difference between deep nodes and tips? Figure 5 shows lots of blue and red dots (Figure 5 needs a legend – what do the colors mean and what metric is being plotted?), meaning overall there does not seem to be a forcing mechanism across mammals for increasing complexity = passive evolution. But this might not be in the case at nodes vs. tips or at subclade level. These data need to be unpicked.

- Conclusions: As mentioned above, the conclusion does not really get into what’s governing the results/patterns presented but rather presents a literature review of what others have found and ideas proposed for vertebral variability (morphology and count). Here I was looking for the ‘ah ha!’ moment when everything would come together into a tight evolutionary story. For instance, one conclusion is that there is non-negligible evidence for a driven trend in complexity but many of the results seem to point to overall passive trends e.g., similar mean values, increase and decrease rate shifts, similar skewness. Irrespective, the authors never really interrogate what might underlie passive/driven trends in the vertebral count of mammals – there is so mechanistic link for the patterns recovered (e.g., what seems to be linked with decreases or increases in count complexity? Why do some clades add thoracics and others lumbaris? What evolutionary phenomenon does your work reveal?).

*****END*****

Author Rebuttal to Initial comments

We thank our referees for giving us an opportunity to revise our original manuscript submission and for their useful and constructive remarks. We address all their points and append our comments, highlighted in blue, next to each. We hope that they find our response satisfactory, and we remain at their disposal for any clarifications.

Reviewer #1 (Remarks to the Author):

This manuscript seeks to address macroevolutionary patterns of complexity in the vertebral column of mammals. The authors test the long-held idea that complexity increases via a driven process (as an intrinsic evolutionary law) as opposed to via random or passive processes. This is an interesting topic that has received much attention in terms of theory, but few tests with empirical data. Therefore, I think this study makes a nice contribution, with a nice large sample, and very appropriate methods. The paper is very well written, and generally requires little improvement. The methods are appropriate, thorough, and well explained.

We thank Reviewer 1, Dr Katrina Jones, for her kind words and very supportive remarks.

Below I provide some comments that I think could improve the paper, but these are mostly to improve readability. I look forward to seeing this published.

Introduction –

- It would really help readers who are not familiar with vertebrae to include an introductory figure with a vertebral column indicating how your metrics were calculated (indices, ratios). I think the paper will appeal to a broad audience, so they might not be familiar with the anatomy. Plus, given the paper is all about vertebrae, it would be nice to see at least one vertebral column! Is low evenness an indicator of higher complexity – the polarity is not obvious to me with this?

We agree on all points. We have included a brand-new figure (Figure 1) illustrating two columns reconstructed from 3D scans, and with colour coded presacral formulae. For each, we include values of the Brillouin and evenness indices of the presacral region, as well as the thoracic:lumbar ratio. We would like to keep the explanations of the index formulae in the Methods. It is true that the polarity is not obvious and we add clarifications in the Methods. Basically, if we adopt an operational definition of complexity, both information theory indices can be considered to capture slightly different aspects of regionalization. In the context of our study both increasing numerical diversity and increasing proportional distribution of vertebrae across column regions act as proxies for “complexity”.

- At the end of the introduction, it would be helpful to clearly state the hypotheses you are testing. Without this, due to the number of analyses, it can seem like a bit of a laundry list of analyses without being clear how each relates to a specific question or hypothesis.

We have included a short section at the end of the introduction where we explain our aims and list our hypotheses.

8Results/discussion

- P6 – you mention that many distributions are bimodal. Why do you think this is? Does this suggest a need for more specific subgroup analyses (ie. Capturing two distinct patterns)?

We are conducting separate studies in this respect. We have not carried out all possible analyses on subclades as our main focus is on general patterns of complexity change. Nonetheless, we highlight some examples of discontinuous distributions in selected clades, and provide a narrative around this.

- It is not clear to me the benefit of calculating the indices on both the presacral and TL regions. Cervical counts are fixed at 7 in mammals (except for rare exceptions), so we would not expect this region to contribute to variation. They seem to show very similar results. It might make the paper more streamlined if you only present one of these in the main text, and move the other to supplement.

We agree and we have selected some plots based upon the thoracolumbar region for the main article and left others in Supplementary Information.

- P7, l156 – “the presacral column features a substantial degree of numerical heterogeneity and evenness”. Again, I think this is very hard for the reader to imagine. Can you describe an example to illustrate or put it in the figure above? E.g., a column in which the proportion of vertebrae in each region is similar. Including these more descriptive statements will really help with interpretation.

In addition to a figure illustrating vertebral columns and index calculations, we have added further notes in the section on indices in Methods. These describe some hypothetical examples that we hope will guide the reader in interpreting the information theory indices. We have eliminated any reference to “heterogeneity”, given that its former use may have engendered confusion.

- P8 – why might cetaceans vary T and L counts in similar proportions? Think its worth mentioning that vertebral morphology is far more homogeneous in cetacea than terrestrial mammals, suggesting less functional differentiation, and perhaps an absence of distinct selection pressures that would cause the ratio to vary.

We devote a substantial part of the Discussion to Cetacea (and some other clades) and we fully agree with (and develop) the remark about distinct selection pressures.

- P9, I210 – Any more interpretation around this pattern? Does this suggest that vertebral columns tend to elongate via the disproportionate addition of thoracics, particularly in shorter columns.

Indeed, we find that this is the case, and we comment upon the disproportionate addition of thoracic elements in the part of the Discussion devoted to Afrotheria and Xenarthra, where the pattern is most obvious.

- P13 – Is it correct in assuming that the contribution of SCW reflects the relative importance of the driven trends i.e., complexity increases are approximately 2/3 passive and 1/3 driven?

Yes, this is broadly correct. The subclade test is very powerful, but nonetheless requires careful interpretation.

Conclusion

- P15 – “We uncover modest but non negligible evidence of driven trends towards increasing column regionalization and evenness.” – I don’t understand what you mean by regionalization here, surely regionalization would be the opposite of evenness? Do you mean a driven trend in T/L, in which case that would seem to be an increase in relative thoracic length ie preferential addition of thoracics. Would increasing evenness mean decreasing complexity?

We have rephrased this part of the text to improve clarity. Furthermore, we have added more comments in the relevant section of the Results. “Regionalization” was a misleading word, so we have removed it.

Figures

- Fig 2 – add key or heading to each figure. It has been shown that this default rainbow color ramp is can be misleading and isn’t color blind friendly. Consider using alternatives in r e.g., vidiris. There are a lot of similar contmaps here – can you move any to supplement e.g. TL vs presacral?

We have redone the continuous trait mapping and chose the 'turbo' colour scale in the *viridis* package. Unfortunately, other scales were less clear. We moved some colour maps to Supplements, as per recommendation.

- Fig 4 – again – do we need both presacral and TL in main text (they are pretty much identical)?

We have simplified and split the figures.

- Fig 5 – remove species names as they are too small to read. Instead adding silhouettes and key node names as in figure one would be more helpful.

Done.

Katrina Jones

Reviewer #2 (Remarks to the Author):

These researchers are interested in evolutionary complexity and chose regional numbers of vertebrae to test the hypothesis of conservation of presacral numbers of vertebrae in mammals. As such, they tackle major issues in evolutionary biology and find interesting results, namely that presacral numbers of vertebrae, and thoracolumbar numbers specifically, are not conserved interspecifically and across clades in mammals as they are broadly thought to be. They do so using phylogenetic analyses and various indices that quantify count data. I am favorable to their approach and the methods employed seem appropriate and creative. The breadth of the study is impressive, as are the beautiful but nearly unwieldy figures. I see this as a welcome contribution but have one major question/concern/request regarding data, then a series of minor edits and suggestions that are listed below that, indicated by Line number (L).

We are grateful to the Reviewer for their encouraging words. We hope that, in the revised version, analyses are used with greater clarity and plots are more streamlined.

Methods: The authors state that they scour google scholar for vertebral counts and synopses from the journal Mammalian Species (MS). Are these two separate processes or one in the same (i.e., are the authors getting these data directly from Mammalian Species and nowhere else)? Generally, I like the idea of sticking a primary source (even if it's a journal), and I wonder if each individual reference paper needs

11to be cited. If these are separate processes and data are coming from any manuscript with the terms searched for, I think these certainly need to be credited and cited somewhere. I do not see a list of these or separate bibliography for them anywhere in the supplementary materials. This is why I assume the primary source is MS. But that could be incorrect and it could be that the authors still should cite the individual papers the data come from even if from MS.

We have added a tabulation of references, as requested. It was not always possible to narrow down formulae to specific specimens, so we have omitted any reference to registration numbers. Also, we have rewritten this part of the Methods because the primary literature amply covers the first-hand data gathering (in addition, specimens on display did not always have accession numbers on them, or at least they were not clearly visible or readily accessible). We hope this is satisfactory.

Methods: It is also stated that one author collected data using first-hand gathering at two museums. Looking over the supplementary materials, I cannot tell which species are represented by published records and which were gathered in novel data collection at museums. It would be important to know what percentage was gathered using each process, if there was overlap (e.g., a record was found for *Canis lupus* and the MR also counted vertebrae in a specimen), and, ideally, the species would be coded according to where the data came from (e.g., published record, museum data, mixed, or the like). It would also be extremely valuable (although perhaps not desired or even practical here) to publish the raw data; that is, specimen numbers and their associated vertebra counts (see Sanchez-Villagra et al., 2007).

Indeed, practicality was an issue here, given time constraints. To simplify things, we have used only primary references. Note: all the comments that follow have been addressed but we have rewritten several sections, and so some of the original text and wording is now changed in the new version.

L110: Add comma between “vertebrae” and n1.

Done.

L235: Should Evenness be capitalized or not? It is capitalized here and in some other places but not in others (e.g., Ls319, 519).

All uppercase removed

L328: Old World should be capitalized and does not require a hyphen (although it can be used).

12Done.

L341: Something is missing from the part of the sentence with “we remark.”

Corrected.

L393-394: The refer to both colugos and tree shrews as gliding taxa. Colugos are gliders, but tree shrews are not; rather, they are scansorial quadrupeds. Additionally, they do not have longer thoracic columns and shorter lumbar columns (with the exception of Ptilocercus, which demonstrates this pattern only slightly).

This section has been removed.

L434: Wording is mixed up and/or something is missing from this sentence around “may seldom alter significantly way the.”

Corrected.

Reviewer #3 (Remarks to the Author):

In this work, Li et al. aim to examine the evolution of vertebral count in mammals by compiling a large species sample size (>1000) and using various measures of complexity: two from information theory (often applied to species diversity) including Brillouin index (“numerical diversity”) and the related Evenness index (“numerical distribution”), while the other two related to thoracic+lumbar counts and T/L ratios. These complexity metrics are then explored using (1) maximum-likelihood ancestral state reconstruction, (2) count-ratio correlations, (3) evolution of vertebral count complexity values using ancestor-descendant analyses over time, (4) passive vs. driven evolution via a subclade (skewness) test, and (5) evolutionary rate shifts across the mammal phylogeny.

The primary conclusions from this study are that the number of presacral vertebrae across mammals is variable, the mean complexity values across major clades do not significantly differ, count is positively related to increases in diversity and evenness but negatively related to T/L ratios. Further, ancestor-descendant relationship shows an overall trend to increasing complexity but that this trend is mainly passive (“with a non-negligible driven trend”). Finally, there are shifts towards both increasing and

13decreasing complexity across the mammalian tree.

Overall, this study was jam packed with data and took an interesting spin on analyzing vertebral count evolution making the foundation work novel. I appreciate the hard work of collecting and analyzing such a large dataset and commend the authors for presenting such a large phylogenetic study using a large suite of methods.

We thank the Reviewer for their several and very helpful suggestions. Much of the paper has been rewritten and restructured as a consequence. We have made nearly all of the requested additions, with the exception of an exhaustive discussion of the patterns associated with all of the major mammal groups. However, we comment briefly on some of the most interesting patterns in the Discussion. We hope that this is satisfactory

However, the study, for me, falls a bit flat as there is little interpretation of the results presented – what does this all mean for mammal (vertebral) evolution or complexity more broadly? The Results & Discussion section is primarily just a summary of the results with little interpretation and the Conclusions is more of a broad literature review as well as discussion on how the next study will include fossils with some preliminary observations. While the Conclusions section does bring up prior work on aspects of ecology, development, biomechanics etc. that may influence vertebral morphology and count, these aspects are not tied back to the current study

Each of those sections has been extensively rewritten and many have been simplified relative to the original versions. We realize there are many aspects of complexity that are dealt with under separate sections, but we think this is necessary in order to provide a comprehensive summary of the observed patterns.

and the data presented. As currently presented, it is hard to see what the data mean in terms of mammalian (vertebral) evolution or the evolution of organismal complexity – what is underlying the count variability presented and the patterns recovered? In sum, the data are great, but the execution of the manuscript has not quite reached the level necessary to provide insight into the process of evolution.

We have provided some explanations for some of the patterns emerging from the results and we have rewritten and amplified the Discussion to accommodate them. We agree that much more could be explained but space constraints and the nature of the study posed limits. We are now in the process of examining ecological determinants of vertebral numerical diversity as part of a separate study.

- General comment: A major conclusion of this study is that it overturns long held ideas that mammals

14have fixed vertebral counts. However, this study is not the first to suggest variable counts (outside the cervical region) in mammals. Rodents have often been considered the least variable with respect to vertebral counts (that 19 TL mentioned), but various other groups have been shown to be much more variable in count including Afrotheres, Xenarthrans, Cetaceans, and some Primates. These groups also pop out as variable in the present analysis. There have also been a number of studies correlating variable counts across mammalian phylogenies with different locomotor modes (like suspensory locomotion or cursorial locomotion – both recent studies and referenced by the authors). Because of this, it feels a bit too strong to state emphatically that this study overturns long held ideas – but it does support recent thought and data on the column being much more variable than previously acknowledged.

We agree and we have rewritten this section of the text. We provide a succinct summary of the most recent findings from published literature and this summary informs a preamble to our discussion. Originally this was part of the Introduction, but in order to keep the latter more succinct we reordered the narrative.

- Introduction: When I read the Introduction, I couldn't help but be reminded of the recent study by Jones and colleagues (2019, NatComms) on the evolution of vertebral complexity in mammals using information theory. Their study was a spinoff of McShea (1993, Evolution) that aimed to use information theory to examine mammal vertebral complexity in select ancestor-descendant crown mammal pairs; he found no evidence for a driven trend. The Jones paper specifically modeled various hypotheses for the evolution of complexity across fossil and extant synapsids as a whole using evolutionary modeling techniques and found evidence for stepwise evolution driven by the release of a function constraint and selection for increased metabolic rate – debunking passive or long-term trends of complexity during the early stages of synapsid and crown mammal evolution. While these two papers are cited in the present work, there is no specific discussion of them in the Introduction (and very little in the Discussion), considering their direct relevance – even if they used morphological complexity instead of count complexity.

We agree, and we have amended the relevant part of the text, which now features at the beginning of the Discussion in a substantially rewritten format. We note that Jones et al. (2019) focus particularly on the transition to the crown using morphometric data, whereas we use a very significantly larger extant taxon sample and metrics of regionalisation. We think that these two studies are complementary and tackle rather different questions.

- Mammal tree. This section does not seem like result or discussion. Perhaps some of this can be worked into the end of the Introduction.

Done.

- Clades – general comment: Some of the clades here are very large and contain many subclades with diverse ecological and locomotor habits. It would be nice to dig into these subclades more to better understand what might be driving some of the patterns recovered. Related to this, Supplementary Dataset 1 does not separate major clades by subclade so it is challenging to trace which groups might be driving any specific patterns.

The referee is absolutely correct. An in-depth analysis of patterns related to individual clades would be a valuable addition. We have included some expansion of this theme, chiefly in the revised Discussion. We realize this may not be what the Reviewer was hoping for. However, we feel the paper would grow enormously if we included detailed commentaries on group-specific patterns. We will be addressing these elsewhere as part of ongoing collaborative projects.

- Complexity shows marked within-group variation. I found this section very hard to follow. One of the major problems is Figure 2 in that it lacks legends and does not give any indication of what the colors mean. In some instances, statements about groups being “high” and “intermediate” etc all looked the same. The color palette is just too hard to follow (note, Red is often used for high or fast and Blue for low or slow). Also, I needed to go back and forth to Figure 1 to figure out what node or clade I was looking at. There needs to be some guideposts if you are going to draw out specific group comparisons.

We have simplified the text and we have included silhouettes of various groups to guide the reader through the elaborate patterns of rate shifts. We tried to create plots with a colour scale, but the results were difficult to interpret. However, the accompanying figure caption should help in that respect. Very simply:

Branches coloured in grey are those showing background rates.

Branches coloured in red (if any are present) are those in which rates are HIGHER than the background rates. The darker the red tone, the higher the rates relative to the grey background rates.

Branches coloured in blue (if any are present) are those in which rates are LOWER than the background rates. The darker the blue tone, the lower the rates relative to the grey background rates.

The circles (pie charts) represent the posterior probabilities of shifts.

The darker the red tone of the circle, the higher the upturn in rate (i.e. the higher the shifts towards an increase).

The darker the blue tone of the circle, the higher the downturn in rate (i.e. the higher the shifts towards a decrease).

The colour of the circle marks the direction of a shift, namely an increase (red tones) or a decrease (blue tones) relative to surrounding adjacent branches

16This section also (and Figure 2 caption) referred to “heterogeneity” for the Brillouin index instead of “numerical diversity” as presented in the Introduction. The violine plots of within and between group variation feels like it should be in the main paper as it provides an important summary of the data collected (why are monotremes and marsupials lumped together here when they are vastly different clades?) and I kept going back and forth to confirm observations. Finally, while indices vary between clades they all have similar means – this is one criteria often used for passive evolution (increases and decreases equally likely).

We have removed any reference to “heterogeneity” throughout the paper. We agree with the Reviewer that the violin plots are best placed in the main text and we have created a new main figure for them. Our grouping of monotremes and marsupials reflects our small sample sizes for both of these non-placental groups (inevitably so in the case of the former). We could easily treat these as two separate clades, but their depauperate sampling means we would effectively have to remove them both from further consideration. Our solution of lumping was therefore a compromise given the nature of our data.

The similar means in complexity indices do not imply that complexity evolved according to a passive diffusion model. In fact, we have tests of increases and decreases in Supplementary Table 4 and these show that increases significantly outnumber decreases for all indices and, furthermore, that the magnitude of the former is significantly greater than the absolute magnitude of the latter.

- Vertebral numbers differ significantly between groups. Post-hoc tests recover differences but how many pair-wise tests were done? Put another way, what percentage of your pair-wise tests were different and were there any specific groups governing these trends?

A detailed breakdown of all comparisons is shown in Supplementary Table 1 and a brief commentary now appears in the main text.

I notice that Afrotheria and Cerartiodactyla – both groups with extensive ecological variation – seem to dominate. With respect to count and ratio correlations, IC of T/L and T+L only explains about 2% of the variance in the data and the data are very scattered. Should you draw a line through this? To me, the only tight relationship here is between T/L and lumbar count, the other two are variable and driven by certain clades.

Indeed we cannot use standard regression lines here and this is why we opted for loess regression curves. We discuss which clades drive which patterns in the now revised Figure 5.

- Complexity trends are underpinned by passive and driven processes. The first part of this section generally repeats the methods – can it be summarized more succinctly? What are your interpretations for the ancestor-descendant plots?. Is complexity increasing over time or is it random? The general results are stated but these are not interpreted in any meaningful way. In terms of skewness, it seems that the overriding trend is passive with some “non-negligible” driven force (the table of results has some odd values for Logit T/L). If there is a driven trend, then what is creating the “hard bound” if anything? If it is passive, why? This test is for the whole dataset but are you able to do similar tests with different ancestral- descendant populations? For instance, do some groups evolve passively while others via a driven trend? Perhaps this might allow you to interpret your data from physiological, biomechanical, or developmental perspective?

We agree. We have carried out analyses for each group (Supplementary Table 4) but we have not undertaken analyses of specific subclades within those groups. We propose to carry out a series of targeted studies for several clades, as part of ongoing research. For the purpose of this paper, however, we are interested in overall patterns across phylogeny and we have selected some major groups to provide a preliminary characterization of changes in serial complexity. An in-depth treatment of individual clades requires additional data collection. However, we devote a small section of the Discussion to describing some of the patterns that we uncovered, especially those that have broad ecological and evolutionary implications.

- Rate shifts: The results show widespread rate shifts across the tree – both increases and decreases. In McShea (1993) he states “if a forcing mechanism has operated, the expectation is that complexity would have increased in evolutionary lineages more frequently than it decreased.” What do you see in your data? Is there a difference between deep nodes and tips? Figure 5 shows lots of blue and red dots (Figure 5 needs a legend – what do the colors mean and what metric is being plotted?), meaning overall there does not seem to be a forcing mechanism across mammals for increasing complexity = passive evolution. But this might not be in the case at nodes vs. tips or at subclade level. These data need to be unpicked.

Rates and trends are distinct concepts and rate shifts may occur even in the absence of a trend. We have explored increases and decreases in values of complexity indices and the results are reported in Supplementary Table 4, where we find evidence that increasing values of complexity are more frequent than decreasing values. In the section on shifts, however increases and decreases relate to rates of change (so either a speeding up or a slowing down in change per unit time), not to increments in the absolute values of indices along branches. We had originally considered the terms “upturns” and “downturns” to refer to rate increases and decreases, but we were concerned that had the potential for

18misreading. As for trends, not only do we find increases in complexity over time (both across phylogeny and in individual groups); we also find evidence that initial complexity values tend to favour downstream increases in complexity along descendant branches.

- Conclusions: As mentioned above, the conclusion does not really get into what's governing the results/patterns presented but rather presents a literature review of what others have found and ideas proposed for vertebral variability (morphology and count). Here I was looking for the 'ah ha!' moment when everything would come together into a tight evolutionary story. For instance, one conclusion is that there is non-negligible evidence for a driven trend in complexity but many of the results seem to point to overall passive trends e.g., similar mean values, increase and decrease rate shifts, similar skewness. Irrespective, the authors never really interrogate what might underlie passive/driven trends in the vertebral count of mammals – there is so mechanistic link for the patterns recovered (e.g., what seems to be linked with decreases or increases in count complexity? Why do some clades add thoracics and others lumbar? What evolutionary phenomenon does your work reveal?).

We have elaborated upon some of these patterns in the Discussion, with emphasis on clades that show the most extreme examples of thoracolumbar differentiation, and we have provided potential explanations that merit further testing.

Decision Letter, first revision:

22nd June 2022

Dear Dr Ruta,

Your manuscript entitled "Divergent vertebral formulae shape the evolution of complexity in mammals" has now been seen by the same three reviewers, whose comments are attached. As you will see from the reports, while reviewer 1 now signs off, reviewers 2 and 3 still have quite a few outstanding comments. Some of these concerns follow through from the point-by-point response to the main manuscript, and may be resolvable through further clarification and discussion. I wanted to draw your attention to reviewer 3's comments in particular, which as they note are interrelated. While we appreciate that this manuscript can't accommodate every sub-clade analysis necessary to thoroughly discuss trends, it's clear from reviewer 3's report that some compromise needs to be made here to enable a substantive followthrough from the questions laid out in the abstract and introduction and the conclusions. Should you be able to resolve this, I am hopeful that we will be able to accept the manuscript for publication.

19Finally, before we invite you to submit a revision and a response to the reviewer comments, I just want to mention that reviewer 2's issue with figure 5 appears to have been browser specific (I can load it without problem) but we will need you to make your data and code available without request both for the purposes of review and on publication.

We therefore invite you to revise your manuscript taking into account all reviewer and editor comments. Please highlight all changes in the manuscript text file [OPTIONAL: in Microsoft Word format].

- * If you have not done so already please begin to revise your manuscript so that it conforms to our Article format instructions at <http://www.nature.com/natecolevol/info/final-submission>. Refer also to any guidelines provided in this letter.

[REDACTED]

Nature Ecology & Evolution is committed to improving transparency in authorship. As part of our efforts in this direction, we are now requesting that all authors identified as 'corresponding author' on published papers create and link their Open Researcher and Contributor Identifier (ORCID) with their account on the Manuscript Tracking System (MTS), prior to acceptance. ORCID helps the scientific community achieve unambiguous attribution of all scholarly contributions. You can create and link

20your ORCID from the home page of the MTS by clicking on 'Modify my Springer Nature account'. For more information please visit www.springernature.com/orcid.

[REDACTED]

Reviewer expertise:

as before

Reviewers' comments:

Reviewer #1 (Remarks to the Author):

I am happy with the changes the authors have made.

Thanks,
Katrina

Reviewer #2 (Remarks to the Author):

Li et al. have revised their manuscript based on reviewer comments. I have reviewed both their rebuttal and revised manuscript. I have a number of relatively minor comments and a few larger concerns, largely relate to inconsistencies between what the authors say in the rebuttal versus what is in the manuscript. First, the authors state that they removed reference to both CTL and TL since they are repetitive, but I still see both referenced in the manuscript.

Second, the authors have put together a data set file with references to the primary literature. However, the methods still state that google searches are utilized in addition to Mammalian Species and other primary literature. The authors could clarify or revise what they did here. I would also suggest citing some of the primary references in the main text in this first part of the Methods section (following "tabulations in the published literature"). I realize citing all of them may not be possible due to reference number restrictions, but citing those that are relied heavily upon for data (Asher et al., Sanchez-Villagra et al., Williams et al., etc.) would be appropriate. Please also include reference to the data file here.

Finally, I cannot view Figure 5 so therefore cannot review it. Other similar plots seems fine, so I assume this one is too but wanted to flag my inability to download or preview it. The remainder of my

21comments are listed by line (L) number:

L27: "non-negligible influence from driven processes" comes out of the blue in the Abstract and is a rather confusing phrase to process without additional context. Can the authors briefly define or explain driven processes here?

L42-47: "is a fundamental component"... "but remains"  "are fundamental components"... "but remain"

L68: othergroups  other groups. Also, what other groups/types of groups are the authors talking about? It is not clear here.

L148-149: Can the authors elaborate briefly here what is meant by null hypothesis 3 "rates of change do not depart from a global optimum value" as many readers will not know what is meant here and how it relates to the bigger picture.

L263: When the authors refer to Marsupialia, do they mean both Marsupialia and Monotremata since they combined those two groups due to small sample sizes of the latter? If so, perhaps use "Marsupialia/Monotremata" throughout.

L432-433: What do the authors mean by "From the above, it follows that..." It is not at all clear to me why this is included when the authors could just state that T:L is negatively correlated with L.

L479: Why is present capitalized here and elsewhere (L1007)?

L608: Does "flying, ground, and tree squirrels" refer to all of Sciuromorpha just certain squirrels? Also, by flying squirrels, are the authors referring to pteromyins (actual sciurids) or anomylurids (scaly-tailed flying squirrels)? If the latter, this should be reworded because it currently appears that the authors are referring only to the sciurids by listing them this way.

L761: "moderately elongate and compact thoracic regions" - elongate makes sense, but what is meant by "compact"? Are the authors referring to the size or close association of the vertebrae in manatees and golden moles? If so, please clarify by explaining and potentially including citation/s or remove compact.

L769: Something is missing from "represented by the appear uniform" - elephant shrews that?

Reviewer #3 (Remarks to the Author):

I would like to commend the authors for rethinking and rewriting how their data are presented and interpreted. The discussion does a much better job of linking the results with broader scale patterns. Specially, the authors now note how some of the most extreme examples of vertebral counts and complexity are associate with clades with highly diverse (or divergent) ecologies/functions (e.g., cetaceans, xenarthrans, bats) and noted developmental differences (although, here

22Afrotheria/Xenarthra should be noted too).

For me, there are still two, interrelated things missing:

(1) Trends/rates: The Introduction sets up the study within the context of trends, and the results detail various analyses, but the Discussion never circles back to this.

“Line 108: Here we test for significant macroevolutionary trends in the complexity of the mammalian presacral column and determine whether any such trends arose by passive/diffusive or driven processes.”

While the new Discussion does a much better job of linking vertebral counts/complexity to clades with disparate ecologies, it is still currently lacking discussion of trends, as well as rates of evolution – both important components of the results. While I realize there are length constraints, the authors fail to get back to the purpose of the paper as set up in the Introduction and Abstract. In fact, while the Discussion now focuses on ecology/function the Abstract summarizes the study wrt to trends with no mention of ecology. Right now there is a disconnect between the Intro and Discussion – it would be nice to bring up the NULL hypotheses as now laid out in the Intro in the Discussion so the study can be balanced.

(2) Passive/driven trends: I realize the authors are very much against doing subclade tests as they want to keep such data for subsequent work (as detailed in the rebuttal letter), but I think that is really where the meat and potatoes is regarding ‘trends’. At the moment, the trend analyses are ‘non-negligible’ probably because the data are being swamped out by including such large clades. The authors have now provided an example (line 327) of how running similar analyses on subclades provides much more nuanced insights (although the data are not provided) but that is where things end. The new Discussion is strongly alluding to ecology/function(development) as being a major driver of vertebral count/complexity (and presumably because the indexes are not related to taxonomic diversity and there are numerous instances of convergence) by pulling out unique clades but this is not supported by any subclade test for understanding macro patterns.

Please see the commented PDF for some line specific comments. Also, please note, if you have developed unique code for this project, it should be made available with the publication and not via permissions from the authors.

[note from the editor--the annotated pdf from reviewer 3 was too big to attach, please download it from <https://acrobat.adobe.com/link/review?uri=urn:aaid:scds:US:50942caa-3230-3b5f-a537-604a214ff912> and contact Luíseach if you have any problems)

*****END*****

Author Rebuttal, first revision:

We thank our referees for allowing us to carry out a new revision of our work on mammalian axial complexity and for their constructive criticism and encouraging comments. We address all their points and append our comments, highlighted in blue, next to each. We hope that they find our response satisfactory, and we remain at their disposal for any further clarifications.

Reviewers' comments:

Reviewer #1 (Remarks to the Author):

I am happy with the changes the authors have made.

Thanks,
Katrina

We thank Dr Katrina Jones for her continuous support through the review process.

Reviewer #2 (Remarks to the Author):

Li et al. have revised their manuscript based on reviewer comments. I have reviewed both their rebuttal and revised manuscript. I have a number of relatively minor comments and a few larger concerns, largely relate to inconsistencies between what the authors say in the rebuttal versus what is in the manuscript.

We thank Reviewer 2 for their suggestions, all of which were endorsed.

First, the authors state that they removed reference to both CTL and TL since they are repetitive, but I still see both referenced in the manuscript.

The corresponding author accepts full responsibility for the confusion engendered in the previous set of remarks. It is true that we intended to focus, for the most part, on thoracolumbar

24patterns. We have kept comments on presacral patterns brief and consigned most information to plots and tabulations in the supplements (e.g., count data; complexity indices). However, especially for analyses of rates shift, we have embedded some remarks on the presacral region where to highlight a few interesting patterns.

Second, the authors have put together a data set file with references to the primary literature. However, the methods still state that google searches are utilized in addition to Mammalian Species and other primary literature. The authors could clarify or revise what they did here.

We have substantially restructured this part of the text. As one can imagine, the retrieval of information on vertebral formulae could not follow a simple and linear pattern. A core set of data gleaned from extensive compendia gave impetus for additional online searches. In some cases, we were fortunate to stumble across monographs with dozens of entries. In other cases, we had to assemble data through various combinations of key words. The monographs in the Mammalian Species series, where available, kept appearing in many of the online searches. Unfortunately, not all synoptic treatments included formulae.

I would also suggest citing some of the primary references in the main text in this first part of the Methods section (following "tabulations in the published literature"). I realize citing all of them may not be possible due to reference number restrictions, but citing those that are relied heavily upon for data (Asher et al., Sanchez-Villagra et al., Williams et al., etc.) would be appropriate. Please also include reference to the data file here.

As requested by the reviewer, we have cited several of the most comprehensive references that provided the main data sources for the full data tabulation. We have also included a reference to Supplementary Dataset 2.

Finally, I cannot view Figure 5 so therefore cannot review it. Other similar plots seems fine, so I assume this one is too but wanted to flag my inability to download or preview it.

The Editor has alerted us to the issue with figure 5, which appears to have been browser specific, and informed us that she can load the figure in question without problem. We have included code for reproducing said figure.

The remainder of my comments are listed by line (L) number:

L27: "non-negligible influence from driven processes" comes out of the blue in the Abstract and is a rather confusing phrase to process without additional context. Can the authors briefly define or explain driven processes here?

The entire Abstract has been rewritten and hopefully the concept of driven trends emerges more clearly

L42-47: "is a fundamental component"... "but remains"  "are fundamental components"... "but remain"

We have reworded this section and added a small chunk of text to convey that complexity is the variable of macroevolutionary dynamics that remains understudied, unlike diversity and disparity which are much better studied and understood and have featured more prominently in macroevolutionary analyses.

L68: othergroups  other groups. Also, what other groups/types of groups are the authors talking about? It is not clear here.

We corrected the typo and cited a few examples of additional groups where serial structures could be – and, in some cases, have been – explored in terms of their complexity. We have shifted the reference numbers after the cited examples, for ease of exposition.

L148-149: Can the authors elaborate briefly here what is meant by null hypothesis 3 "rates of change do not depart from a global optimum value" as many readers will not know what is meant here and how it relates to the bigger picture.

We have rephrased the relevant section for clarity to make it more accessible. We have eliminated 'global optimum' from the new version

L263: When the authors refer to Marsupialia, do they mean both Marsupialia and Monotremata since they combined those two groups due to small sample sizes of the latter? If so, perhaps use "Marsupialia/Monotremata" throughout.

We have amended the relevant uses of the word as per reviewer's suggestion. Indeed, in most cases, the use of the term implies that we are referring to Monotremata and Marsupialia combined, and we clarify instances where this is not the case.

L432-433: What do the authors mean by "From the above, it follows that..." It is not at all clear to me why this is included when the authors could just state that T:L is negatively correlated with L.

As per reviewer's suggestion, we have removed the phrase in question and started the new sentence in the way that the referee suggests.

L479: Why is present capitalized here and elsewhere (L1007)?

This is now corrected throughout so that the word 'present' is in lower case.

L608: Does "flying, ground, and tree squirrels" refer to all of Sciuromorpha just certain squirrels? Also, by flying squirrels, are the authors referring to pteromyins (actual sciurids) or anomylurids (scaly-tailed flying squirrels)? If the latter, this should be reworded because it currently appears that the authors are referring only to the sciurids by listing them this way.

We have rectified this by specifying that we are indeed referring to sciurids.

L761: "moderately elongate and compact thoracic regions" - elongate makes sense, but what is

meant by "compact"? Are the authors referring to the size or close association of the vertebrae in manatees and golden moles? If so, please clarify by explaining and potentially including citation/s or remove compact.

We have opted to remove the word 'compact'.

L769: Something is missing from "represented by the appear uniform" - elephant shrews that?

We have eliminated the incorrect phrase and reorganized the sentences around it, clarifying that we are indeed talking about elephant shrews.

Reviewer #3 (Remarks to the Author):

I would like to commend the authors for rethinking and rewriting how their data are presented and interpreted. The discussion does a much better job of linking the results with broader scale patterns. Specially, the authors now note how some of the most extreme examples of vertebral counts and complexity are associate with clades with highly diverse (or divergent) ecologies/functions (e.g., cetaceans, xenarthrans, bats) and noted developmental differences (although, here Afrotheria/Xenarthra should be noted too).

We thank Reviewer 3 for their comments and annotated pdf. We are very pleased to learn that they liked the previous Discussion and we certainly hope that the second revision does a little more justice to the evolutionary patterns that we uncover. It is difficult to refer to each annotation on the old version of the text, not least because the new text is substantially revised and expanded. However, with the new revision all of the stylistic glitches from the previous text should be fixed. We decided to eliminate most of the content that was formerly part of the discussion on fossil data.

For me, there are still two, interrelated things missing:

(1) Trends/rates: The Introduction sets up the study within the context of trends, and the results detail various analyses, but the Discussion never circles back to this.

We are convinced by this. Hopefully, the new and expanded version is less 'dry' than the previous one. We have indeed circled back to our three null hypotheses and we add a commentary to each (under separate headers) in the new text. Part of the difficulty in this process was that, originally, Results and Discussion were intermingled. We now believe, in agreement with Reviewer 3, that a separate discussion works better. Now, both trends and rates re-emerge, hopefully bringing more balance to the whole work.

“Line 108: Here we test for significant macroevolutionary trends in the complexity of the mammalian presacral column and determine whether any such trends arose by passive/diffusive or driven processes.”

While the new Discussion does a much better job of linking vertebral counts/complexity to clades with disparate ecologies, it is still currently lacking discussion of trends, as well as rates of evolution – both important components of the results. While I realize there are length constraints, the authors fail to get back to the purpose of the paper as set up in the Introduction and Abstract. In fact, while the Discussion now focuses on ecology/function the Abstract summarizes the study wrt to trends with no mention of ecology. Right now there is a disconnect between the Intro and Discussion – it would be nice to bring up the NULL hypotheses as now laid out in the Intro in the Discussion so the study can be balanced.

We think (hope) that Introduction and Discussion are now linked more directly and a more cohesive narrative emerges. The null hypotheses now feature in the revised Discussion

(2) Passive/driven trends: I realize the authors are very much against doing subclade tests as they want to keep such data for subsequent work (as detailed in the rebuttal letter), but I think that is really where the meat and potatoes is regarding 'trends'. At the moment, the trend analyses are 'non-negligible' probably because the data are being swamped out by including such large clades. The authors have now provided an example (line 327) of how running similar analyses on subclades provides much more nuanced insights (although the data are not provided) but that is where things end. The new Discussion is strongly alluding to ecology/function (development) as being a major driver of vertebral count/complexity (and presumably because the indexes are not related to taxonomic diversity and there are numerous instances of convergence) by pulling out unique clades but this is not supported by any subclade test for understanding macro patterns.

We have included the required tests, and produced a new figure (Fig. 7) to illustrate the main findings. We take the liberty, respectfully, of anticipating a possible remark by the Reviewer about Glires which, despite being the most diverse clade, was not subjected to a subclade test. The corresponding author takes full responsibility for this choice, and for the selection of the five groups that now feature in the main results devoted to the skewness partition. Glires gave us a substantial headache and, due to time constraints, we decided to focus on more 'manageable' clades. We hope this is satisfactory and we thank the Reviewer for convincing us to attempt to see 'the woods from the trees', so to speak,

Please see the commented PDF for some line specific comments. Also, please note, if you have developed unique code for this project, it should be made available with the publication and not via permissions from the authors.

R code is now available in the supplementary material.

[note from the editor--the annotated pdf from reviewer 3 was too big to attach, please download it from <https://acrobat.adobe.com/link/review?uri=urn:aaid:scds:US:50942caa-3230-3b5f-a537-604a214ff912> and contact Luíseach if you have any problems)

Decision Letter, second revision:

17th October 2022

Dear Dr. Ruta,

Thank you for submitting your revised manuscript "Divergent vertebral formulae shape the evolution of complexity in mammals" (NATECOLEVOL-220115698B). It has now been seen again by the original reviewers and their comments are below. The reviewers find that the paper has improved in revision, and therefore we'll be happy in principle to publish it in Nature Ecology & Evolution, pending minor revisions to satisfy the reviewers' final requests and to comply with our editorial and formatting guidelines.

We are now performing detailed checks on your paper and will send you a checklist detailing our

30editorial and formatting requirements in about a week. Please do not upload the final materials and make any revisions until you receive this additional information from us.

[REDACTED]

Reviewer #2 (Remarks to the Author):

The authors have revised the manuscript to my satisfaction. I only have one comment that needs to be addressed: On line 411, the authors refer to "two monophyletic groups, the Old World and New World monkeys." Old World monkeys are monophyletic, but I suspect that the authors mean catarrhines (OW monkeys and apes) here. Unless the authors excluded hominoids from analyses (and I don't think they did and would be concerned if they did), this wording must be changed. What the authors are referring to could be words as "anthropoids" or if they want to be more specific, "anthropoids (platyrrhines and catarrhines)." Writing out the common nomenclature is also an option, but it becomes a bit more burdensome: "anthropoids (New World monkeys and Old World monkeys and apes)" or the like. The point of all this is that hominoids (apes and humans) are not OW monkeys, but they are all catarrhines.

Reviewer #3 (Remarks to the Author):

I would like to commend the authors for this much improved manuscript - the work is clearly expressed, the abstract/intro and discussion are more tightly linked, and the potential underlying factors driving trends in the data more clearly development (as well as identifying limitations). I understand this was a lot of work, but I think this version of the manuscript is much stronger and will have greater impact upon publication. Thank you and congratulations!

Our ref: NATECOLEVOL-220115698B

27th October 2022

Dear Dr. Ruta,

Thank you for your patience as we've prepared the guidelines for final submission of your Nature

31Ecology & Evolution manuscript, "Divergent vertebral formulae shape the evolution of complexity in mammals" (NATECOLEVOL-220115698B). Please carefully follow the step-by-step instructions provided in the attached file, and add a response in each row of the table to indicate the changes that you have made. Please also check and comment on any additional marked-up edits we have proposed within the text. Ensuring that each point is addressed will help to ensure that your revised manuscript can be swiftly handed over to our production team.

****We would like to start working on your revised paper, with all of the requested files and forms, as soon as possible (preferably within two weeks). Please get in contact with us immediately if you anticipate it taking more than two weeks to submit these revised files.****

In recognition of the time and expertise our reviewers provide to Nature Ecology & Evolution's editorial process, we would like to formally acknowledge their contribution to the external peer review of your manuscript entitled "Divergent vertebral formulae shape the evolution of complexity in mammals". For those reviewers who give their assent, we will be publishing their names alongside the published article.

Nature Ecology & Evolution offers a Transparent Peer Review option for new original research manuscripts submitted after December 1st, 2019. As part of this initiative, we encourage our authors to support increased transparency into the peer review process by agreeing to have the reviewer comments, author rebuttal letters, and editorial decision letters published as a Supplementary item. When you submit your final files please clearly state in your cover letter whether or not you would like to participate in this initiative. Please note that failure to state your preference will result in delays in accepting your manuscript for publication.

Cover suggestions

As you prepare your final files we encourage you to consider whether you have any images or illustrations that may be appropriate for use on the cover of Nature Ecology & Evolution.

Nature Ecology & Evolution has now transitioned to a unified Rights Collection system which will allow our Author Services team to quickly and easily collect the rights and permissions required to publish your work. Approximately 10 days after your paper is formally accepted, you will receive an email in providing you with a link to complete the grant of rights. If your paper is eligible for Open Access, our Author Services team will also be in touch regarding any additional information that may be required to arrange payment for your article.

Please note that *Nature Ecology & Evolution* is a Transformative Journal (TJ). Authors may publish their research with us through the traditional subscription access route or make their paper immediately open access through payment of an article-processing charge (APC). Authors will not be required to make a final decision about access to their article until it has been accepted. [Find out more about Transformative Journals](https://www.springernature.com/gp/open-research/transformative-journals)

Authors may need to take specific actions to achieve [compliance with funder and institutional open access mandates](https://www.springernature.com/gp/open-research/funding/policy-compliance-faqs). If your research is supported by a funder that requires immediate open access (e.g. according to [Plan S principles](https://www.springernature.com/gp/open-research/plan-s-compliance)) then you should select the gold OA route, and we will direct you to the compliant route where possible. For authors selecting the subscription publication route, the journal's standard licensing terms will need to be accepted, including [those licensing terms](https://www.nature.com/nature-portfolio/editorial-policies/self-archiving-and-license-to-publish) will supersede any other terms that the author or any third party may assert apply to any version of the manuscript.

Please use the following link for uploading these materials:
[REDACTED]

[REDACTED]

Reviewer #2:

Remarks to the Author:

The authors have revised the manuscript to my satisfaction. I only have one comment that needs to be addressed: On line 411, the authors refer to "two monophyletic groups, the Old World and New World monkeys." Old World monkeys are monophyletic, but I suspect that the authors mean catarrhines (OW monkeys and apes) here. Unless the authors excluded hominoids from analyses (and I don't think they did and would be concerned if they did), this wording must be changed. What the authors are referring to could be words as "anthropoids" or if they want to be more specific, "anthropoids (platyrrhines and catarrhines)." Writing out the common nomenclature is also an option, but it becomes a bit more burdensome: "anthropoids (New World monkeys and Old World monkeys and apes)" or the like. The point of all this is that hominoids (apes and humans) are not OW monkeys, but they are all catarrhines.

Reviewer #3:

Remarks to the Author:

I would like to commend the authors for this much improved manuscript - the work is clearly expressed, the abstract/intro and discussion are more tightly linked, and the potential underlying factors driving trends in the data more clearly development (as well as identifying limitations). I understand this was a lot of work, but I think this version of the manuscript is much stronger and will have greater impact upon publication. Thank you and congratulations!

Author Rebuttal, second revision:

We thank Reviewers 2 and 3 for their additional sets of comments, all of which we have endorsed

#2 (Remarks to the Author):

The authors have revised the manuscript to my satisfaction. I only have one comment that needs to be addressed: On line 411, the authors refer to "two monophyletic groups, the Old World and New World monkeys." Old World monkeys are monophyletic, but I suspect that the authors mean catarrhines (OW monkeys and apes) here. Unless the authors excluded

34hominoids from analyses (and I don't think they did and would be concerned if they did), this wording must be changed. What the authors are referring to could be words as "anthropoids" or if they want to be more specific, "anthropoids (platyrrhines and catarrhines)." Writing out the common nomenclature is also an option, but it becomes a bit more burdensome: "anthropoids (New World monkeys and Old World monkeys and apes)" or the like. The point of all this is that hominoids (apes and humans) are not OW monkeys, but they are all catarrhines.

Response: We thank the reviewer for highlighting our incorrect use of the phrase 'Old World monkeys' and for suggesting alternative phrases. We opted for the simplest solution and chose 'Old World monkeys and apes' in the relevant section of the text. That section is now folded into Supplementary Information.

Reviewer #3 (Remarks to the Author):

I would like to commend the authors for this much improved manuscript - the work is clearly expressed, the abstract/intro and discussion are more tightly linked, and the potential underlying factors driving trends in the data more clearly development (as well as identifying limitations). I understand this was a lot of work, but I think this version of the manuscript is much stronger and will have greater impact upon publication. Thank you and congratulations!

Response: We thank the reviewer for their enthusiastic support.

Final Decision Letter:

3rd January 2023

Dear Marcello,

Happy New Year.

We are pleased to inform you that your Article entitled "Divergent vertebral formulae shape the evolution of anatomical complexity in mammals", has now been accepted for publication in Nature Ecology & Evolution.

35Over the next few weeks, your paper will be copyedited to ensure that it conforms to Nature Ecology and Evolution style. Once your paper is typeset, you will receive an email with a link to choose the appropriate publishing options for your paper and our Author Services team will be in touch regarding any additional information that may be required

You will not receive your proofs until the publishing agreement has been received through our system

Due to the importance of these deadlines, we ask you please us know now whether you will be difficult to contact over the next month. If this is the case, we ask you provide us with the contact information (email, phone and fax) of someone who will be able to check the proofs on your behalf, and who will be available to address any last-minute problems . Once your paper has been scheduled for online publication, the Nature press office will be in touch to confirm the details.

Acceptance of your manuscript is conditional on all authors' agreement with our publication policies (see www.nature.com/authors/policies/index.html). In particular your manuscript must not be published elsewhere and there must be no announcement of the work to any media outlet until the publication date (the day on which it is uploaded onto our web site).

Please note that *Nature Ecology & Evolution* is a Transformative Journal (TJ). Authors may publish their research with us through the traditional subscription access route or make their paper immediately open access through payment of an article-processing charge (APC). Authors will not be required to make a final decision about access to their article until it has been accepted. [Find out more about Transformative Journals](https://www.springernature.com/gp/open-research/transformative-journals)

Authors may need to take specific actions to achieve [compliance with funder and institutional open access mandates](https://www.springernature.com/gp/open-research/funding/policy-compliance-faqs). If your research is supported by a funder that requires immediate open access (e.g. according to [Plan S principles](https://www.springernature.com/gp/open-research/plan-s-compliance)) then you should select the gold OA route, and we will direct you to the compliant route where possible. For authors selecting the subscription publication route, the journal's standard licensing terms will need to be accepted, including [self-archiving-and-license-to-publish](https://www.nature.com/nature-portfolio/editorial-policies/self-archiving-and-license-to-publish). Those licensing terms will supersede any other terms that the author or any third party may assert apply to any version of the manuscript.

36If you have any questions about our publishing options, costs, Open Access requirements, or our legal forms, please contact ASJournals@springernature.com

We welcome the submission of potential cover material (including a short caption of around 40 words) related to your manuscript; suggestions should be sent to Nature Ecology & Evolution as electronic files (the image should be 300 dpi at 210 x 297 mm in either TIFF or JPEG format). Please note that such pictures should be selected more for their aesthetic appeal than for their scientific content, and that colour images work better than black and white or grayscale images. Please do not try to design a cover with the Nature Ecology & Evolution logo etc., and please do not submit composites of images related to your work. I am sure you will understand that we cannot make any promise as to whether any of your suggestions might be selected for the cover of the journal.

You can generate the link yourself when you receive your article DOI by entering it here: <http://authors.springernature.com/share>.

[REDACTED]

P.S. Click on the following link if you would like to recommend Nature Ecology & Evolution to your librarian <http://www.nature.com/subscriptions/recommend.html#forms>

** Visit the Springer Nature Editorial and Publishing website at http://editorial-jobs.springernature.com?utm_source=ejp_NEcoE_email&utm_medium=ejp_NEcoE_email&utm_campaign=ejp_NEcoE for more information about our career opportunities. If you have any questions please click [here](mailto:editorial.publishing.jobs@springernature.com).**